# FROM GRAPHS TO HYPERGRAPHS: HYPERGRAPH PROJECTION AND ITS RECONSTRUCTION

**Yanbang Wang, Jon Kleinberg**
Department of Computer Science, Cornell University
`{ywangdr,kleinberg}@cs.cornell.edu`

## ABSTRACT

We study the implications of the modeling choice to use a graph, instead of a hypergraph, to represent real-world interconnected systems whose constituent relationships are of higher order by nature. Such a modeling choice typically involves an underlying projection process that maps the original hypergraph onto a graph, and is common in graph-based analysis. While hypergraph projection can potentially lead to loss of higher-order relations, there exists very limited studies on the consequences of doing so, as well as its remediation. This work fills this gap by doing two things: (1) we develop analysis based on graph and set theory, showing two ubiquitous patterns of hyperedges that are root to structural information loss in all hypergraph projections; we also quantify the combinatorial impossibility of recovering the lost higher-order structures if no extra help is provided; (2) we still seek to recover the lost higher-order structures in hypergraph projection, and in light of (1)'s findings we propose to relax the problem into a learning-based setting. Under this setting, we develop a learning-based hypergraph reconstruction method based on an important statistic of hyperedge distributions that we find. Our reconstruction method is evaluated on 8 real-world datasets under different settings, and exhibits consistently good performance. We also demonstrate benefits of the reconstructed hypergraphs via use cases of protein rankings and link predictions.

## 1 INTRODUCTION

Graphs are an abstraction of many real-world complex systems, recording pairs of related entities by nodes and edges. Hypergraphs further this idea by extending edges from node pairs to node sets of arbitrary sizes, admitting a more expressive form of encoding for higher-order relationships.

In graph-based analysis, a fundamental and enduring issue exists with the modeling choice of using a graph, instead of a hypergraph, to represent complex systems whose constituent relationships are of higher order by nature. For example, coauthorship networks and social networks, as popular subjects in graph-based analysis, often use edges to denote two-author collaborations or two-person conversations, respectively. Protein interaction networks do the same for protein co-occurrence in biological processes. However, coauthorships, conversations, and biological processes all typically involve more than just two authors, people, and proteins. Reviewing more concrete cases in previous research, we find that this issue usually arises in one of the following two scenarios:

- **"Unobservable":** In some key scenarios, the most available technology for data collection can only detect pairwise relations, as is common in social science and biological science. For example, in the study of social interactions (Madan et al., 2011; Ozella et al., 2021; Dai et al., 2020), sensing methodologies to record physical proximity can be used to build networks of face-to-face interaction: an interaction between two people is recorded by thresholding distances between their body-worn sensors. There is no way to directly record the multiple participants in each conversation by sensors only. In the study of protein interactions (Li et al., 2010; Yu & Kong, 2022; Spirin & Mirny, 2003), detecting protein components of a multiprotein complex all at once poses way more technical barriers than identifying pairs of interacting proteins. Methods for the latter are often regarded as being more economic, high-throughput, reliable, and thus more commonly used.

- **"Unpublished":** Even when they are technically observable, in practice the source hypergraph datasets of many studies are never released. For example, many influential studies analyzing coauthorships do not make available a hypergraph version of their dataset (Newman, 2004; Sarigöl et al., 2014). Many graph benchmarks, including arXiv-hepth (Leskovec et al., 2005), ogbl-collab (Hu et al., 2020), and ogbn-product(Hu et al., 2020), also do not provide their hypergraph originals. Yet the underlying, unobserved hypergraphs contain important information about the domain.

In both cases above, there is often an underlying **hypergraph projection** process that maps the original hypergraph onto a graph on the same set of nodes, and that each hyperedge in the hypergraph is mapped to a clique (i.e. a complete subset where all nodes are pairwise connected) in the graph. In other words, two nodes are connected in the projected graph iff they coexist within a hyperedge.

While it's easy to perceive the loss of some higher-order relations during hypergraph projection, to this date we still have many crucial unanswered questions regarding this issue's detailed cause, consequences, and potential remediation: **(Q1)** what connection patterns of hyperedges in the original hypergraph are combinatorially impossible to recover after the projection? **(Q2)** What are the theoretical worst cases that these connection patterns can create, and how frequent do they occur in real-world hypergraph datasets? **(Q3)** Given a projected graph, is it possible to reconstruct a hypergraph out of it if some reasonable extra help is allowed, and if so, how? **(Q4)** How might the reconstructed hypergraph offer advantages over the projected graph?

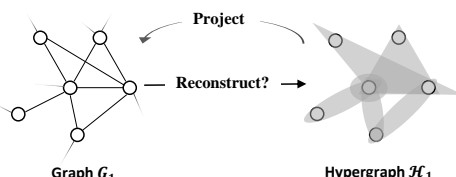

**Hypergraph reconsruction.** We note that all the questions above essentially point to a common problem structure which is the **reversal of the hypergraph projection** process: there is an underlying hypergraph $\mathcal{H}_1$ that we can't directly observe; instead we can only access its *projected graph* (or *projection*) $G_1$. Our goal is to reconstruct $\mathcal{H}_1$ as accurately as possible from $G_1$. The first two questions above assume no extra help (input) should be given in the reconstruction, other than the projected graph $G_1$ itself. The latter two questions permit some extra input, which we will elaborate in Sec.4.

Figure 1: hypergraph reconstruction as the reversal of hypergraph projection. Given projected graph $G_1$, the goal is to reconstruct the original hypergraph $\mathcal{H}_1$ in real world as accurately as possible.

For broad applicability, we assume no multiplicities for $G_1$'s edges: they just say whether two nodes co-exist in at least one hyperedge. Appx. F.9 addresses the simpler case with edge multiplicities.

**Previous work.** Very limited work investigated implications of hypergraph projection or its reversal (*i.e.* hypergraph reconstruction). For implications, the only work to our knowledge is Wolf et al. (2016), which compares hypergraph and its projected graph for computational efficiency on spectral clustering; the former was found to be more efficient. For reconstruction, Young et al. (2021) is by far the closest, which however studies how to use *least* number of cliques to cover the projected graph, whose principle does not really apply to real-world hypergraphs (see experiment in Sec.5).

In graph mining, two relevant problems are hyperedge prediction and community detection, yet both are still vastly different in setup. For hyperedge prediction (Benson et al., 2018a;b; Yadati et al., 2020; Xu et al., 2013), its input is a hypergraph, rather than a projected graph. Methods for hyperedge prediction also only identify hyperedges from a given set of candidates, rather than the large implicit spaces of all possible hyperedges. Community detection, on the other hand, looks for densely connected regions of a graph under various definitions, but not for hyperedges. Both tasks are very different from the goal of searching and inferring hyperedges over the projected edges. We include discussion of other related work in Appendix C.

## 2 PRELIMINARIES

**Hypergraph**. A hypergraph $\mathcal{H}$ is a tuple $(V, \mathcal{E})$: $V$ is a set of nodes, and $\mathcal{E} = \{E_1, E_2, ..., E_m\}$ is a set of sets with $E_i \subseteq V$ for all $1 \le i \le m$. For the purpose of reconstruction, we assume the hyperedges are distinct, *i.e.* $\forall 1 \le i, j \le m, E_i \ne E_j$.

**Projected Graph.** $\mathcal{H}$'s *projected graph (i.e. projection, clique expansion)*, $G$, is a graph with the same node set $V$, and (undirected) edge set $\mathcal{E}'$, *i.e.* $G = (V, \mathcal{E}')$, where two nodes are joined by an edge in $\mathcal{E}'$ **iff** they belong to a common hyperedge in $\mathcal{E}$. That is, $\mathcal{E}' = \{(v_i, v_j)|v_i, v_j \in E, E \in \mathcal{E}\}$.

**Maximal Cliques.** A *clique* $C$ is a fully connected subgraph. We use $C$ to also denote the set of nodes in the clique. A *maximal clique* is a clique that cannot become larger by including more nodes. The *maximal clique algorithm* returns all maximal cliques $\mathcal{M}$ in a graph, and its time complexity is linear to $|\mathcal{M}|$ (Tomita et al., 2006). A *maximum clique* is the largest maximal clique in a graph.

## 3 ANALYSIS OF HYPERGRAPH PROJECTION AND RECONSTRUCTION

We first analyze hypergraph projection and its reversal from the lens of graph theory and set theory, addressing **(Q1)** and **(Q2)** raised above.

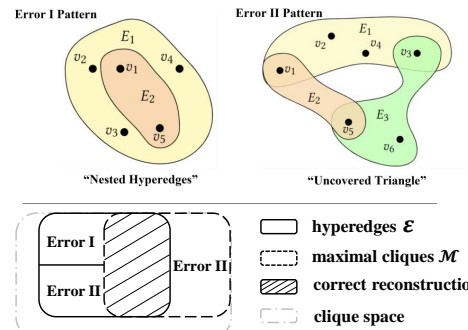

| Dataset | $|\mathcal{E}|$ | $|\mathcal{E}'|$ | $|\mathcal{M}|$ | Err.I | Err.II |
|---|---|---|---|---|---|
| DBLP (Benson et al., 2018a) | 197,067 | 194,598 | 166,571 | 2.02% | 18.9% |
| Enron (Benson et al., 2018a) | 756 | 300 | 362 | 42.5% | 53.3% |
| Foursquare (Young et al., 2021) | 1,019 | 874 | 8,135 | 1.74% | 88.6% |
| Hosts-Virus(Young et al., 2021) | 218 | 126 | 361 | 19.5% | 58.1% |
| H. School (Benson et al., 2018a) | 3,909 | 2864 | 3,279 | 14.9% | 82.7% |

Figure 2: The upper panel shows hyperedge patterns that trigger errors in max-clique based reconstruction. Error I leads to missing $E_2$ (false negative), while Error II results in an incorrect identification of max clique $v_1, v_3, v_5$ as a hyperedge (false positive) and overlooks $v_1, v_5$ (false negative). The lower panel shows the errors' associations with $\mathcal{E}$ and $\mathcal{M}$.

Table 1: $\mathcal{E}$ is the set of hyperedges; $\mathcal{E}'$ is the set of hyperedges not nested in any other hyperedges; $\mathcal{M}$ is the set of maximal cliques in $G$. **Error I, II** result from the violation of Conditions I, II, respectively. Error I = $\frac{|\mathcal{E} \backslash \mathcal{E}'|}{|\mathcal{E} \cup \mathcal{M}|}$, Error II = $\frac{|\mathcal{M} \backslash \mathcal{E}'| + |\mathcal{E}' \backslash \mathcal{M}|}{|\mathcal{E} \cup \mathcal{M}|}$. Errors caused by violation of both conditions are counted as Error I.

**Hyperedge patterns that are hard to recover after projection (Q1)**

In principle, any clique in a projection can be a true hyperedge. Therefore, toward perfect reconstruction we should consider $\mathcal{U}$, the universe of all cliques in $G$, including single nodes. To enumerate $\mathcal{U}$, a helpful view is the union of all maximal clique's power set: $\mathcal{U} = \bigcup_{C \in \mathcal{M}} \mathcal{P}(C) \smallsetminus \varnothing$. In that sense, maximal clique algorithm is a critical first step for hypergraph reconstruction, as also applied by Young et al. (2021) to initialize its MCMC solver. In the extreme case, it's easy to see that if $\mathcal{H}$'s hyperedges are all disjoint, $G$'s maximal cliques would be exactly $\mathcal{H}$'s hyperedges. It is impossible to find all hyperedges without finding all maximal cliques. Therefore, we consider the reconstruction accuracy of maximal clique algorithm a good measure of the reconstruction's difficulty.

**Theorem 1.** *The maximal cliques of $G$ are exactly all hyperedges of $\mathcal{H}$, i.e. $\mathcal{M} = \mathcal{E}$, if and only if the following **two conditions** hold:*

    **I.** *for every hyperedge $E \in \mathcal{E}$ there does not exist a hyperedge $E' \in \mathcal{E}$ s.t. $E \subset E'$;*

    **II.** *every maximal clique in $G$ is a hyperedge in $\mathcal{H}$, i.e. $\mathcal{M} \subseteq \mathcal{E}$.*

Theorem 1 gives the two necessary and sufficient conditions that characterize when $\mathcal{H}$ is "easy" to reconstruct. Note that Condition I is the famous Sperner property (Greene & Kleitman, 1976), and Condition II is the definition of "conformal" as in Colomb & Nourine (2008). In terms of their implications on hyperedge patterns, Condition I is quite self-explanatory, which simply forbids the pattern of "nested" hyperedges. In comparison, Condition II is much less intelligible. Our following theorem further interprets Condition II by the hyperedge pattern of "uncovered triangle".

**Theorem 2.** *A hypergraph $\mathcal{H} = (V, \mathcal{E})$ is conformal **iff** for every three hyperedges there always exists some hyperedge $E$ such that all pairwise intersections of the three hyperedges are subsets of $E$, i.e.:*
$$\forall E_i, E_j, E_q \in \mathcal{E}, \ \exists E \in \mathcal{E}, \ s.t. \ (E_i \cap E_j) \cup (E_j \cap E_q) \cup (E_q \cap E_i) \subseteq E$$

Theorem 2 shows that for a hypergraph to satisfy Condition II, it can't allow any three hyperedges in itself to form a "triangle" whose three vertices are "uncovered". Intuitively, the triangle in this pattern would induce a 3-clique among the "vertices" of the triangle, which would not be part of a hyperedge if the triangle is uncovered. The value of Theorem 2 is that it gives a nontrivial interpretation of Condition II in Theorem 1 by showing how to check a hypergraph's conformity just based on its hyperedge patterns, without computing any maximal cliques. Based on the two conditions, we can now further define the two types of errors made by maximal cliques.

**Definition 1.** *Every error made due to treating all maximal cliques in $\mathcal{G}$ as true hyperedges in $\mathcal{H}$ can be attributed to $\mathcal{H}$'s violation of at least one of the conditions in Theorem 1. An error is defined to be **Error I (Error II)** if it is caused by the $\mathcal{H}$'s violation of Condition I (Condition II).*

Fig.2 illustrates the two error-triggering hyperedge patterns (*i.e.* "error patterns" for brevity hereafter) corresponding to the two errors, as well as their relationship to other important concepts. Also note that here the Errors I & II are different from (but related to) the well-known Type I (false positive) and Type II (false negative) errors in statistics. See Appx. B.5 for more discussion.

**Empirical frequency of the hyperedge patterns, and their theoretical worst case (Q2)**

Both error patterns in Fig.2 have simple construct, so it is perceivable that they are common in

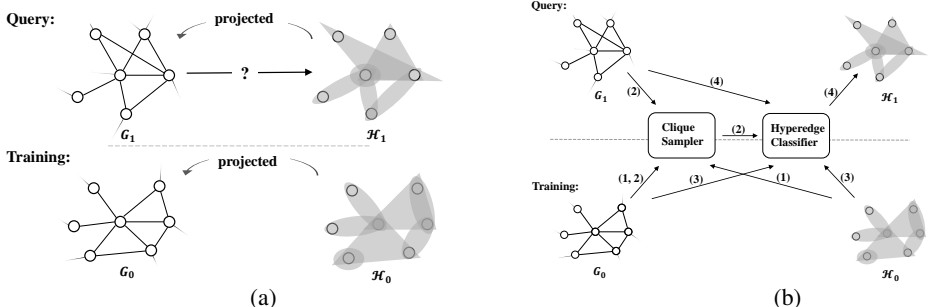

Figure 3: **(a)** Supervised hypergraph reconstruction. $\mathcal{H}_0$ and $\mathcal{H}_1$ belong to the same application domain. Given $\mathcal{H}_0$ (and its projection $G_0$), the task is to reconstruct $\mathcal{H}_1$ from its projection $G_1$. **(b)** 4-step reconstruction: (1) the clique sampler is optimized on $G_0$ and $\mathcal{H}_0$; (2) the clique sampler samples candidates from $G_0$ and $G_1$, then passes result to the hyperedge classifier; (3) the hyperedge classifier extracts features of candidates from $G_0$ and trains on them; (4) the hyperedge classifier extracts features of candidates from $G_1$ and identify hyperedges.

real-world hypergraphs. Table 1shows the frequency of the error patterns and their resulted error rate in reconstruction. We see that Error I patterns are common in hypergraphs of emails and social interactions. In the worst case, a hypergraph contains one large hyperedge and many nested hyperedges as proper subsets.

That said, one can argue that there may be many real-world hypergraphs that (almost) satisfy Condition I. It turns out that the Error II patterns caused by violating Condition II is also disastrous.

**Theorem 3.** *Let $\mathcal{H} = (V, \mathcal{E})$ be a hypergraph that only satisfies Condition I in Theorem 1, with $m = |\mathcal{E}|$. Denote by $p^{(\mathcal{H})}$ the accuracy of maximal clique algorithm for reconstructing $\mathcal{H}$. Then,*

$$min_{\mathcal{H}} \; p^{(\mathcal{H})} \leq 2^{-\binom{m-1}{\lceil m/2 \rceil - 1}} \ll 2^{-m}$$

Theorem 3 shows that our Error II pattern can also give rise to (super-)exponentially many hyperedges that are almost impossible to be combinatorially recovered, if we only rely on the projected graph.

## 4 LEARNING-BASED HYPERGRAPH RECONSTRUCTION

**Overview**. This section will introduce a new learning-based hypergraph reconstruction paradigm, including the problem setup and the proposed method. The idea is that in addition to the projected graph as input, we also assume to know a hypergraph (or part of it) from the same distribution as the reconstruction target. Sec.4.1 will formulate the problem and explain why this setup is meaningful in practice. Sec.4.2.1 - 4.2.3 will present details of the reconstruction method.

### 4.1 PROBLEM DESCRIPTION

We've identified the inherent difficulty of reconstructing a hypergraph from its projection without additional information. However, as presented at the beginning of this paper, in practice, there are many cases where it would be highly desirable if we can recover the lost higher-order relations from projection. How do we reconcile this theoretical "impossibility" with practical necessities?

The key observations are two. First, the noted "impossibility" pertains specifically to the aim of *flawlessly* reconstructing a hypergraph *solely* based on its projection, especially in theoretical worst case. However, this doesn't preclude our option to *approximate* a hypergraph *within a specific application domain*, particularly if we have insights about the domain's typical hypergraph structures.

Secondly, in the context of a specific application domain with one or more projected graphs to be reconstructed, **collecting just a single "hypergraph sample" (or even a portion of it) within the domain can be immensely beneficial**. This sample aids in understanding the typical structure and patterns of hypergraphs in that particular domain. For instance, to reconstruct the coauthorship hypergraphs from graphs claimed to be built from an earlier DBLP dataset, accessing and learning from a sample of another coauthorship hypergraph, perhaps from DBLP of a more recent time period, or a different database like Microsoft Academic Graph (MAG, Sinha et al. (2015)), can be invaluable.

In addition, resources like human-labeled or crowd-sourced data, such as surveys (Ozella et al., 2021), can also be potential source of the hypergraph sample. In the case of protein interaction networks, expert-provided labels for a distinct yet related species or organ could be particularly helpful —

for example, using one of the following databases to reconstruct the other: Reactome (Croft et al., 2010) and hu.MAP 2.0 (Drew et al., 2017). We also note that the hypergraph sample's size doesn't necessarily need to match that of the reconstruction target. Sec.5 will provide a detailed illustration of all the scenarios mentioned above.

The crucial observations above point us to a relaxed version of the problem, which involves the usage of *another hypergraph from the same application domain* as training data. The new learning-based paradigm is shown Fig.3a, as an update to Fig.1, with the following the problem statement.

**Learning-based hypergraph reconstruction.** **(1) input** $X$: a projected graph $G$; **(2) output** $y$: the original hypergraph $\mathcal{H}$; **(3) split**: as in Fig.3a, $(X_{\text{train}}, y_{\text{train}}) = (G_0, \mathcal{H}_0)$, $(X_{\text{query}}, y_{\text{query}}) = (G_1, \mathcal{H}_1)$, given $\mathcal{H}_0, \mathcal{H}_1 \sim \mathcal{D}$. **(4) metric:** following Young et al. (2021) we use Jaccard score to evaluate reconstruction accuracy: $\frac{|\mathcal{E}_1 \cap \mathcal{R}_1|}{|\mathcal{E}_1 \cup \mathcal{R}_1|}$, where $\mathcal{E}_1$ is the true hyperedges; $\mathcal{R}_1$ is the reconstructed hyperedges.

The reconstructed hypergraphs in this learning-based setting offer two advantages. Most importantly, they still serve to crucially identify interactive node groups in a projected graph, irrespective of whether supervised signals are used. Also, they can potentially enhance various downstream tasks, such as node ranking and link prediction (see Sec.5.4), as they aggregate information from both the projected graph and the application domain.

It's important to note that, in the second advantage above, **our goal of hypergraph reconstruction isn't to outperform SOTA methods in downstream tasks**. In fact, SOTA methods for a specific downstream task are often end-to-end customized, in which cases reconstructed hypergraphs would not be necessary. Rather, we highlight the value of reconstructed hypergraphs as an informative and handy intermediate representation, especially when specific downstream tasks are undetermined at the time of reconstruction, which is similar to the role word embeddings play in language tasks.

## 4.2 Proposed Method for Learning-based Hypergraph Reconstruction

We are now ready to present our method for the problem. Apparently, even with the training hypergraph, the greatest challenge here is still the enormous search space of potential hyperedges (*i.e.* "clique space" in Fig.2). The novel idea here is that we will use a *clique sampler* to narrow the search space of hyperedges, and then use a *hyperedge classifier* to identify hyperedges from the narrowed space. Both modules are optimized using the training data. Fig.3b gives a more detailed 4-step view.

### 4.2.1 $\rho(n, k)$-alignment

We now start by introducing an important statistic that we found, which is the foundation of our proposed method: $\rho(n, k)$ is a statistic that describes distributions of hyperedges inside maximal cliques. Given a hypergraph $\mathcal{H} = (V, \mathcal{E})$, its projection $G$, maximal cliques $\mathcal{M}$: $\rho(n, k)$ is the probability that we find a unique hyperedge by randomly sampling a size-$k$ subset of nodes from an arbitrary size-$n$ maximal clique. A $(n, k)$ is called *valid* if $1 \le k \le n \le N$; $N$ is the *maximum clique's size*. $\rho(n, k)$ can be empirically estimated via the unbiased estimator $\hat{\rho}(n, k)$:

$$\hat{\rho}(n, k) = \frac{|\mathcal{E}_{n,k}|}{|\mathcal{Q}_{n,k}|}, \quad \text{where} \begin{cases} \mathcal{E}_{n,k} & = \{S \in \mathcal{E} | S \subseteq C, |S| = k, C \in \mathcal{M}, |C| = n\} \\ \mathcal{Q}_{n,k} & = \{(S, C) | S \subseteq C, |S| = k, C \in \mathcal{M}, |C| = n\} \end{cases}$$

$\mathcal{E}_{n,k}$ is all size-$k$ hyperedges in size-$n$ maximal cliques; $Q_{n,k} = |\{C | C \in \mathcal{M}, |C| = n\}|\binom{n}{k}$ is all ways to sample a size-$n$ maximal clique and then a size-$k$ subset (*i.e.* $k$-clique) from the maximal clique.

Our key observation is that if two hypergraphs *e.g.* $\mathcal{H}_0, \mathcal{H}_1$, are generated from the same source (application domain), they should have similar $\rho(n, k)$'s, which we call $\rho(n, k)$-*alignment*. Fig.4a uses heatmaps to visualize $\rho(n, k)$-alignment on a famous email dataset, Enron (Benson et al., 2018a), where $\mathcal{H}_0$ and $\mathcal{H}_1$ are split based on a middle timestamp of the emails (hyperedges). $n$ is plot on the $y$-axis, $k$ on the $x$-axis. Fig.4b plots $\rho(n, k)$'s for 5 other datasets. Both figures show the distributions of $\rho(n, k)$ to align well between training and query splits; in contrast, the distributions of $\rho(n, k)$ across datasets are much different. Appendix Fig.17 confirms this visual observation quantitatively.

Also note Fig.4a's second column and diagonal cells are darkest, implying that the $\binom{n}{k}$ term in $Q_{n,k}$ can't dominate $\hat{\rho}(n, k)$: $\binom{n}{k}$ is smallest at $k = n$ or 1, growing exponentially as $k \to n/2$ regardless of data. $\hat{\rho}(n, k)$ peaking at $k = 2$ shows that the data term $|\mathcal{E}_{n,k}|$ plays a numerically meaningful role. **Complexity.** The complexity for computing $\rho(n, k)$ is $O(|\mathcal{M}|)$. See Appx. D.1 for details.

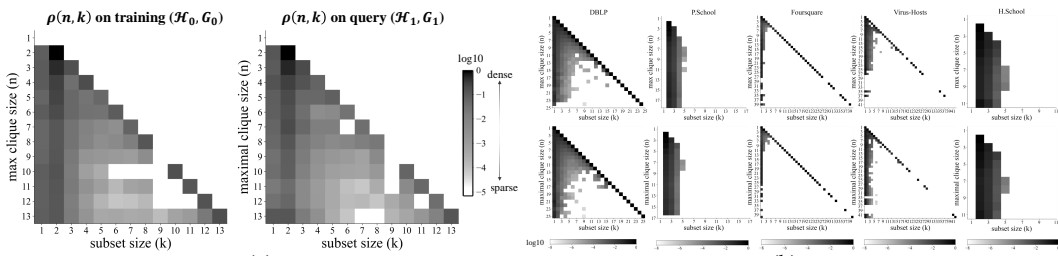

Figure 4: (a) $\rho(n,k)$-alignment on dataset Enron; $\mathcal{H}_0, \mathcal{H}_1$ obtained by splitting all emails by a middle timestamp. (b) $\rho(n,k)$-alignment on more datasets. Notice the column-wise similarity and row-wise difference.

### 4.2.2 CLIQUE SAMPLER

Given a query graph, we can't afford to take all its cliques $\mathcal{U}_1$ as candidates for hyperedges. Therefore, we create a clique sampler. Assume a limited sampling budget $\beta \ll |\mathcal{U}_1|$, our goal is to collect as many hyperedges as possible by sampling $\beta$ cliques from $\mathcal{U}$. Any hyperedge missed in sampling will never get identified by the hyperedge classifier later on, so this step is crucial.

Query $G_1$ is not enough to locate hyperedges in the enormous search space of $\mathcal{U}_1$. Fortunately, we can get hints from $G_0$ and $\mathcal{H}_0$. The idea is that we use $G_0$ and $\mathcal{H}_0$ to optimize a clique sampler that can provably collect many hyperedges. The optimization (Fig.3b step 1) is a process to learn "where to sample". Then in $G_1$, we use the optimized clique sampler to sample cliques (Fig.3b step 2). The clique sampler takes the following form:

($\star$) **For each valid** $(n,k)$**, we sample a total of** $r_{n,k}|Q_{n,k}|$ **size-**$k$ **subsets (*i.e.* $k$-cliques) from size-**$n$ **maximal cliques in the query graph, subject to the sampling budget:** $\sum_{n,k} r_{n,k}|Q_{n,k}| = \beta$**.**

$r_{n,k} \in [0,1]$ is the sampling ratio of the $(n,k)$ cell, $|Q_{n,k}|$ is the size of that cell's sample space in $G_0$. To instantiate a sampler, a $r_{n,k}$ should be specified for every valid $(n,k)$. How to determine the $r_{n,k}$'s? We optimize $r_{n,k}$'s towards collecting the most hyperedges from $G_0$, with the objective:

$$\{r_{n,k}\} = \operatorname*{argmax}_{\{r_{n,k}\}} \mathbb{E}(|\bigcup_{(n,k)} r_{n,k} \odot \mathcal{E}_{n,k}|)$$

$\odot$ is a *set sampling* operator that yields a uniformly downsampled subset of $\mathcal{E}_{n,k}$ at downsampling rate $r_{n,k}$. $\odot$ is essentially a generator for *random finite set* (Mullane et al., 2011) (See Appx. B.4). $\mathbb{E}(|\cdot|)$ is expected cardinality. Given $\rho(n,k)$-alignment and objective fullfilled, the optimized sampler should also collect many hyperedges when applied to $G_1$. See Appx. F.8 for empirical validation.

**Optimization.** To collect more hyperedges from $G_0$, a heuristic is to allocate all budget to the darkest cells of the training data's heamap (Fig.4a-left), where hyperedges most densely populate. However, a caveat is that the set of hyperedges $\mathcal{E}_{n,k}$ in each $(n,k)$ cell of the same column are not disjoint. In other words, a size-$k$ clique can appear in multiple maximal cliques of different sizes.

---

**Algorithm 1** Optimize Clique Sampler

**Require:** $\beta$; $N$; $\mathcal{E}_{n,k}$, $\mathcal{Q}_{n,k}$ for all $1 \le k \le N, k \le n \le N$
1: **for** $k = 1$ to $N$ **do**                                              ▷ traverse $k$ to initialize state variables
2:       $\Gamma_k \leftarrow \varnothing$                      ▷ union of $\mathcal{E}_{n,k}$'s picked from column $k$
3:       $\omega_k \leftarrow \{k, k+1, ..., N\}$              ▷ available column-$k$ cells
4:       $r_{i,k} \leftarrow 0$ for $i \in \omega_k$           ▷ sampling ratios for column-$k$ cells
5:       $\Delta_k, n_k \leftarrow$ **UPDATE** $(k, \omega_k, \Gamma_k, \mathcal{E}_{\cdot,k}, \mathcal{Q}_{\cdot,k})$
6: **end for**
7: **while** $\beta > 0$ **do**                                              ▷ the greedy selection starts
8:       $k \leftarrow \operatorname{argmax}_i \Delta_i$       ▷ selects the next best $k$
9:       $r_{n_k,k} \leftarrow \min\{1, \frac{\beta}{|\mathcal{Q}_{n_k,k}|}\}$    ▷ samples cell by ratio
10:      $\Gamma_k \leftarrow \Gamma_k \cup \mathcal{E}_{n_k,k}$    ▷ updates state variables
11:      $\omega_k \leftarrow \omega_k \backslash \{n_k\}$
12:      $\beta \leftarrow \beta - |\mathcal{Q}_{n_k,k}|$
13:      $\Delta_k, n_k \leftarrow$ **UPDATE** $(k, \omega_k, \Gamma_k, \mathcal{E}_{\cdot,k}, \mathcal{Q}_{\cdot,k})$
14:      **if** $\max_k \Delta_k = 0$ **then** break               ▷ breaks if all cells sampled
15: **end while**
16: **return** $r_{n,k}$ for all $1 \le k \le N, k \le n \le N$

---

Therefore, taking the darkest cells may not give best result. In fact, optimizing the objective above involves maximizing a monotone submodular function under budget constraint, which is NP-hard. In light of this, we design Algorithm 1 to greedily approach the optimal solution with worst-case guarantee. It takes four inputs: sampling budget $\beta$, size of the *maximum* clique $N$, $\mathcal{E}_{n,k}$ and $\mathcal{Q}_{n,k}$ for all $1 \le k \le n \le N$. Lines 1-6 initialize state variables. Lines 7-15 run greedy selection iteratively.

---

**Subroutine 1** UPDATE in Algorithm1

---

**Require:** $k; \omega_k; \Gamma_k; \mathcal{E}_{\cdot,k}; \mathcal{Q}_{\cdot,k}$

  **if** $\omega_k \ne \varnothing$ **then**

    $\Delta' \leftarrow \max_{n \in \omega_k} \frac{|\Gamma_k \cup \mathcal{E}_{n,k}| - |\Gamma_k|}{|\mathcal{Q}_{n,k}|}$

    $n' \leftarrow \arg\max_{n \in \omega_k} \frac{|\Gamma_k \cup \mathcal{E}_{n,k}| - |\Gamma_k|}{|\mathcal{Q}_{n,k}|}$

  **else**

    $\Delta' \leftarrow 0; n' \leftarrow 0;$

  **end if**

  **return** $\Delta', n'$

---

The initialization is done column-wise. In each column $k$, $\Gamma_k$ stores the union of all $\mathcal{E}_{n,k}$ selected from column $k$ so far; $\omega_k$ stores the row indices of all cells in column $k$ that haven't been selected; $r_{i,k}$ is the sampling ratio of each valid cell; line 5 calls the subroutine **UPDATE** to compute $\delta_k$, the best sampling efficiency among all available cells, and $n_k$, the row index of that most efficient cell.

Lines 7-15 run the greedy selection. In each iteration, we take the next most efficient cell among the best of all columns, store the selection in the corresponding $r$, and update $(\Gamma, \omega, \delta_k, n_k)$; $k$ is the column index of the selected cell. Only column $k$ needs to be updated as $\mathcal{E}_{n,k}$'s with different $k$'s are independent. We stop when reaching budget or having traversed all cells.

Appx. F.8 visualizes Algorithm 1's iterations and does ablation studies on its necessity.

**Theorem 4.** *Let $q$ be the expected number of hyperedges in $\mathcal{H}_0$ drawn by the clique sampler optimized by Algo. 1; let $q^*$ be the expected number of hyperedges in $\mathcal{H}_0$ drawn by the best-possible clique sampler, under the same $\beta$. Then, $q > (1 - \frac{1}{e})q^* \approx 0.63q^*$.*

Theorem 4 bounds the optimality of Algo. 1; $\frac{q}{q^*}$ in practice can be much higher than $0.63$. Also notice that Algo. 1 leaves at most one $(n, k)$ cell partially sampled. ***Is that a good design?*** In fact, there always exists an optimal clique sampler that leaves at most one cell partially sampled. Otherwise, we can always relocate all our budget from one partially sampled cell to another to achieve a higher $q$.

Appx.D.3 further discusses Algo. 1's relationship with the standard submodular optimization, how it eliminates Errors I & II, and the tuning of $\beta$ from the perspective of precision-recall tradeoff.

**Complexity.** Algo.1's average complexity is $O(|\mathcal{E}|)$, worst complexity is $O(N|\mathcal{E}|)$. See Appx. D.3.

### 4.2.3 HYPEREDGE CLASSIFIER

A *hyperedge classifier* is a binary classification model that takes a target clique in the projection as input, and outputs a 0/1 label indicating whether the target clique is a hyperedge. We train the hyperedge classifier on $(\mathcal{H}_0, G_0)$ which has ground-truth labels (step 3, Fig.3b), then use it to identify hyperedges from $G_1$ (step 4, Fig.3b). To serve this purpose, a hyperedge classifier should contain two parts (1) a feature extractor that extracts expressive features for characterizing a target clique, and (2) a binary classifier that transforms a feature vector into a 0/1 label. (2) is a standard task, so we use a MLP with 100 hidden neurons. (1) requires more careful design, as discussed below.

**Design Principles.** Creating an effective feature extractor necessitates identifying the essential information about a target clique in the projection. Since our setting doesn't assume attributed nodes or edges, leveraging the structural features both *within* and *around* the target clique is crucial. Also, positional embeddings like Deepwalk are not applicable here due to unaligned node IDs.

In principle, any structural learning model for characterizing connectivities can be a potentially good choice — and there are many of them operating on individual nodes (Henderson et al., 2012; Li et al., 2020; Xu et al., 2018; Yin et al., 2020). However, since this is a new task involving complex clique structures, we want to have a learning model as interpretable as possible in such a"clique-rich" context. Here, we introduce two feature extractors that achieve this via interpretable features, though alternative options exist.

**"Count" Feature Extractor.** Many powerful graph structural learning models use different notions of "count" to characterize local connectivity patterns. For example, GNNs typically use node degrees as initial features when node attributes are unavailable; the Weisfeiler-Lehman Test also updates a node's color based on the count of different colors in its neighborhood.

Viewing a target clique as a subgraph in a projection, the notion of "count" can be especially rich in meaning: a target clique can be characterized by the count of its [own nodes / neighboring nodes / neighboring edges / attached maximal cliques] in many different ways. We create a total of 8 types of

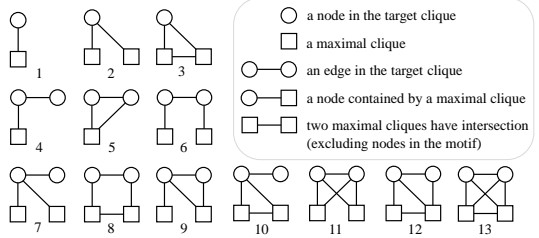

Figure 5: 13 "clique motifs" that encode rich connectivity patterns around a target clique $C$. Each clique motif is formed by [1 or 2 nodes in the target clique] + [1 or 2 maximal cliques that contain the nodes].

Table 2: Summary of the datasets (query split). $\mu(\mathcal{E})$ and $\sigma(\mathcal{E})$ stand for mean and std. of hyperedge's size. $\bar{d}(V)$: average node degrees *w.r.t.* hyperedges. $\mathcal{M}$: maximal cliques. See Table 6 for the full version.

| Dataset | $\|V\|$ | $\|\mathcal{E}\|$ | $\mu(\mathcal{E})$ | $\sigma(\mathcal{E})$ | $\bar{d}(V)$ | $\|\mathcal{M}\|$ |
|---|---|---|---|---|---|---|
| Enron (Benson et al., 2018a) | 142 | 756 | 3.0 | 2.0 | 16 | 362 |
| DBLP (Benson et al., 2018a) | 319,916 | 197,067 | 3.0 | 1.7 | 1.8 | 166,571 |
| P. School (Benson et al., 2018a) | 242 | 6,352 | 2.4 | 0.6 | 64 | 15,017 |
| H. School (Benson et al., 2018a) | 327 | 3,909 | 2.3 | 0.5 | 28 | 3,279 |
| Foursquare (Young et al., 2021) | 2,334 | 1,019 | 6.4 | 6.5 | 2.8 | 8,135 |
| Hosts-Virus (Young et al., 2021) | 466 | 218 | 5.6 | 9.0 | 2.6 | 361 |
| Directors (Young et al., 2021) | 522 | 102 | 5.4 | 2.2 | 1.2 | 102 |
| Crimes (Young et al., 2021) | 510 | 256 | 3.0 | 2.3 | 1.5 | 207 |

generalized count-based features, elaborated in Appx. D.4. Despite technical simplicity, these count features works surprisingly well and can be easily interpreted. See Appx. D.4 also.

**"Motif" Feature Extractor.** As a second attempt , we create a novel "clique motif" feature extractor. we use maximal cliques as intermediaries to bridge the gap between projection and hyperedges. The maximal cliques serve as initial estimations of the high-order structures, encompassing full projection information and offering partially refined insights into higher-order structures. Also, the interaction between maximal cliques and nodes in the target clique form rich connectivity patterns, generalizing the notion of motif (Milo et al., 2002).

Fig.5 lists all 13 connectivity patterns involving the target clique's 1 or 2 nodes and maximal cliques. Clique motifs, compared to count features, more systematically extract structural properties, with two component types (node, maximal clique) and three relation types explained in the legend. A clique motif is *attached* to a target clique $C$ if the clique motif contains at least one node of $C$. We further use $\Phi_i^{(C)}$ to denote the set of type-$i$ ($1 \le i \le 13$) clique motifs attached to $C$.

Given a target clique $C$, how to use clique motifs attached to $C$ to characterize structures around $C$? We define $C$'s structural features as a concatenation of 13 vectors: $[u_1^{(C)}; u_2^{(C)}; ...; u_{13}^{(C)}]$. $u_i^{(C)}$ is a vector of statistics describing the vectorized distribution of type-$i$ clique motifs attached to $C$:

$$u_i^{(C)} = \textbf{SUMMARIZE}(P_i^{(C)}), \qquad P_i^{(C)} = \begin{cases} [\textbf{COUNT}(C, i, \{v\}) \text{ for } v \text{ in } C], & \text{if } 1 \le i \le 3; \\ [\textbf{COUNT}(C, i, \{v1, v2\}) \text{ for } v_1, v_2 \text{ in } C], & \text{if } 4 \le i \le 13; \end{cases}$$

$P_i^{(C)}$ is a vectorized distribution in the form of an array of counts regarding $i$ and $C$. $\textbf{COUNT}(C, i, \chi) = |\{\phi \in \Phi_i^{(C)} | \chi \subseteq \phi\}|$. Finally, $\textbf{SUMMARIZE}(P_i^{(C)})$ is a function that transforms a vectorized distribution $P_i^{(C)}$ into a vector of statistical descriptors. Here we simply define the statistical descriptors as:

$$\textbf{SUMMARIZE}(P_i^{(C)}) = [\textbf{mean}(P_i^{(C)}), \textbf{std}(P_i^{(C)}), \textbf{min}(P_i^{(C)}), \textbf{max}(P_i^{(C)})]$$

As we have 13 clique motifs, these amount to 52 structural features. On the high level, clique motifs extend the well-tested motif methods on graphs to hypergraph projections with clique structures. Compared to count features, clique motifs capture structural features in a more principled manner.

## 5 EXPERIMENTS AND FINDINGS

### 5.1 EXPERIMENTAL SETTINGS

**Baselines.** We adapt 7 methods from four task domains for their best relevance or state-of-the-art performance: (1) Community Detection: Demon (Coscia et al., 2012), CFinder (Palla et al., 2005); (2) Clique Decomposition: MaxClique (Bron & Kerbosch, 1973), Clique Covering (Conte et al., 2016); (3) Hyperedge Prediction: Hyper-SAGNN (Zhang et al., 2019), CMM (Zhang et al., 2018), HPRA (Kumar et al., 2020a); (4) Probabilistic Models: Bayesian-MDL(Young et al., 2021). See more in Appendix C.

**Data & Training.** We use 8 real-world datasets from various application domains. For each dataset, we split the hyperedges to generate $\mathcal{H}_0, \mathcal{H}_1, G_0, G_1$. The properties of $\mathcal{H}_1$ are summarized in Table 2. See Appx. F.1 for more details of dataset split, baselines selection and adaptation, and tuning.

**Reproducibility.** Our code and data are available **here**.

### 5.2 EVALUATING QUALITY OF RECONSTRUCTION

We name our approach **SHyRe** (**S**upervised **Hy**pergraph **Re**construction) whose performance is shown in Table 3. SHyRe variants significantly outperform all baselines in most datasets (7/8), with best improvement in hard datasets such as P. School, H.School, and Enron.This success is attributed

| | DBLP | Enron | P.School | H.School | Foursquare | Hosts-Virus | Directors | Crimes |
|---|---|---|---|---|---|---|---|---|
| CFinder | 11.35 | 0.45 | 0.00 | 0.00 | 0.39 | 5.02 | 41.18 | 6.86 |
| Demon | - | 2.35 | 0.09 | 2.97 | 16.51 | 7.28 | 90.48 | 63.81 |
| Maximal Clique | 79.13 | 4.19 | 0.09 | 2.38 | 9.62 | 22.41 | **100.0** | 78.76 |
| Clique Covering | 73.15 | 6.61 | 1.95 | 6.89 | **79.89** | 41.00 | **100.0** | 75.78 |
| Hyper-SAGNN | 0.13±0.01 | 4.79±0.08 | 12.55±0.33 | 8.96±0.11 | 0.01±0.01 | 7.36±0.50 | 1.97±1.13 | 0.88± 0.17 |
| CMM | 0.11±0.04 | 0.52±0.04 | 19.63±2.74 | 6.28±0.44 | 0.00±0.00 | 4.98± 1.21 | 2.61±0.60 | 0.57±0.29 |
| HPRA | 63.99±3.57 | 10.25±0.42 | 24.32±0.88 | 34.30±2.47 | 47.95±1.33 | 26.51± 0.52 | 73.44±5.60 | 53.16±0.93 |
| Bayesian-MDL | 73.08±0.00 | 4.57±0.07 | 0.18±0.01 | 3.58±0.03 | 69.93±0.59 | 40.24±0.12 | **100.0±0.00** | 74.91±0.11 |
| **SHyRe-count** | **81.18±0.02** | 13.50±0.32 | 42.60±0.61 | **54.56±0.10** | 74.56±0.32 | **48.85±0.11** | **100.0±0.00** | 79.18±0.42 |
| **SHyRe-motif** | **81.19±0.02** | **16.02±0.35** | **43.06±0.77** | 54.39±0.25 | 71.88±0.28 | 45.16±0.55 | **100.0±0.00** | **79.27±0.40** |

Table 3: Performance of hypergraph reconstruction measured in Jaccard score (%, defined in Sec.4.1). Std. is dropped for deterministic methods. "-" means the algorithm did not stop in 72 hours.

| | DBLP | Hosts-Virus | Enron |
|---|---|---|---|
| Best Baseline (full) | 79.13 | 41.00 | 6.61 |
| SHyRe-motif (full) | 81.19±0.02 | 45.16±0.55 | 16.02±0.35 |
| SHyRe-count (20%) | 81.17±0.01 | 44.02±0.39 | 6.43±0.18 |
| SHyRe-motif (20%) | 81.17±0.01 | 44.48±0.21 | 10.56±0.92 |

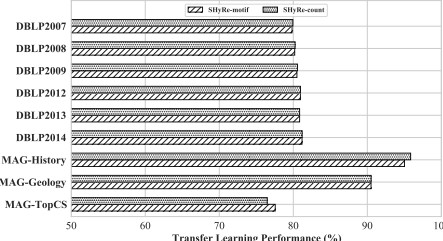

Table 4: Performance of semi-supervised reconstruction using 20% training hyperedges, measured in Jaccard Score.

Figure 6: Transfer learning performance: trained on DBLP2011, tested on various coauthorship datasets.

to SHyRe's innovative use of the training graph and its ability to capture strong, interpretable features, as detailed in Appx. D.4. Although Clique Covering and Bayesian-MDL perform relatively well, they struggle with dense hypergraphs. Hyperedge prediction methods have similar issues despite being also learning-based methods, a topic elaborated in Appx.C.3.

**More metrics.** Fig.17 in Appendix further shows fine-grained performance measured by partitioned errors (Def.1): SHyRe variants significantly reduce more Errors I and II than other baselines do. Besides, we also study various topological properties of the reconstructed hypergraphs and compare them to those of the original hypergraphs. We find that on these new measurements SHyRe variants also produce more faithful reconstructions than baselines. See Appx.F.2 for more details.

### 5.3 SEMI-SUPERVISED LEARNING AND TRANSFER LEARNING

We study more constrained scenarios where we only have access to a small subset of hyperedges, or a a training hypergraph from a nearby domain. These settings correspond to semi-supervised learning and transfer learning. For semi-supervised setting, we choose three datasets of different difficulties: DBLP, Hosts-Virus, and Enron. For each dataset, we randomly discard 80% hyperedges in the training split. Table 4 shows the result: the reconstruction accuracy drops on all datasets, but SHyRe trained on 20% data still outperforms the best baseline on full data.

For transfer learning, we train SHyRe on DBLP 2011, and test on various other DBLP slices and Microsoft Academic Graphs (Sinha et al., 2015), shown in Fig.6. We can see that SHyRe remains robust to the distribution shift as its performance on DBLP 2011 is $81.19\%$ according to Table 3.

### 5.4 USE CASES OF RECONSTRUCTED HYPERGRAPHS IN DOWNSTREAM TASKS

We evaluate advantages of using reconstructed hypergraphs as informative intermediate representations for downstream tasks in comparison to relying on projected graphs, through two use cases: node ranking in protein-protein interaction (PPI) network and link prediction. The details are presented in Appx.F.4 and F.5, respectively. For the former, we apply our method to recover multiprotein complexes from pairwise interaction data, which are then used to rank the essentiality of proteins based on their node degrees. This produces a ranking list that is much better aligned with the ground truth than results based on the projected version of PPI network, as shown in Appendix Table 8.

In the second use case, link prediction, we demonstrate that reconstructed hypergraphs enhance performance over projected graphs across multiple datasets measured by AUC and Recall. These use cases show the efficacy of our hypergraph reconstruction technique in deriving richer, more informative data representations that can benefit downstream analytical tasks.

**More Experiment**. We report more experiment in Appx.F.7-F.12, including ablation studies on clique sampler, optimal sampling coefficients ($r_{m,k}$'s), and running time and storage comparison.

## 6 CONCLUSION

We studied hypergraph projection and its reversal. We identified specific hyperedge patterns that trigger errors, and introduced a novel learning-based approach for hypergraph reconstruction. For future studies, considering different projection expansions like "star" or "line" can be highly promising. We also discuss the **Broader Impacts** of our work in Appx. E.

## ACKNOWLEDGEMENT

We thank Immanuel Trummer for his valuable feedback. This work is supported in part by a Simons Investigator Award, a Vannevar Bush Faculty Fellowship, AFOSR grant FA9550-19-1-0183, and a grant from the MacArthur Foundation.

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

# Appendix

## A REPRODUCIBILITY

Our code and data can be downloaded from `https://anonymous.4open.science/r/supervised_hypergraph_reconstruction-FD0B/README.md`.

## B PROOFS AND ADDITIONAL DISCUSSIONS FOR SEC.3

### B.1 PROOF OF THEOREM 1

*Proof.* **"Only if" direction:** Condition I holds because every hyperedge is a maximal clique, and two maximal cliques cannot be proper subset of each other. Therefore, it is impossible for any two hyperedges $E, E' \in \mathcal{E}$, which are both maximal cliques, to still satisfy $E \subset E'$. Condition II holds trivially because $\mathcal{M} = \mathcal{E} \Rightarrow \mathcal{M} \subseteq \mathcal{E}$.

**"If" direction:** starting from Condition II, it only remains to show that every hyperedge in $\mathcal{H}$ is also a maximal clique in $G$. We prove by contradiction. If there exists a hyperedge $E$ that is not a maximal clique, by definition of maximal clique and because $E$ itself is a clique, $E$ must be a proper subset of some maximal clique $E'$. However, based on Condition II, $E'$ is also a hyperedge. This leads to the relationship $E \in E'$, a contradiction with Condition I. □

### B.2 PROOF OF THEOREM 2

*Proof.* The "if" direction: Suppose that $\mathcal{H}$ is not conformal. According to Def.2, we know that there exists a maximal clique $C \notin \mathcal{E}$. Clearly for every $C$ with $|C| \leq 2$, $|C| \in \mathcal{E}$. For a $C \notin \mathcal{E}$ and $|C| \geq 3$, pick any two nodes $v_1, v_2 \in C$. Because they are connected, $v_1, v_2$ must be in some hyperedge $E_i$. Now pick a third node $v_3 \notin E_i$. Likewise, there exists some different $E_j$ such that $v_1, v_3 \in E_2$, and some different $E_q$ such that $v_2, v_3 \in E_3$. Notice that $E_j \neq E_q$ because otherwise the three nodes would be in the same hyperedge. Now we have $\{v_1, v_2, v_3\} \subseteq (E_i \cap E_j) \cup (E_j \cap E_q) \cup (E_q \cap E_i)$. Because $\{v_1, v_2, v_3\}$ is not in the same hyperedge, $(E_i \cap E_j) \cup (E_j \cap E_q) \cup (E_q \cap E_i)$ is also not in the same hyperedge.

The "only if" direction: Because every two of the three intersections share a common hyperedge, their union is a clique. The clique must be contained by some hyperedge, because otherwise the maximal clique containing the clique is not contained by any hyperedge. □

Alternatively, there is a less intuitive proof that builds upon results from existing work in a detour: It can be proved that $\mathcal{H}$ being conformal is equivalent to its dual $\mathcal{H}'$ being Helly Berge (1973). According to an equivalence to Helly property mentioned in Berge & Duchet (1975), for every set $A$ of 3 nodes in $\mathcal{H}'$, the intersection of the edges $E_i$ with $|E_i \cap A| \geq k$ is non-empty. Upon a dual transformation, this result can be translated into the statement of Theorem 2. We refer the interested readers to the original text.

### B.3 PROOF OF THEOREM 3

Given a set $X = \{1, 2, ..., m\}$ and a hypergraph $\mathcal{H} = (V, \{E_1, ...E_m\})$, we define $f : 2^X \to 2^V$ to be:

$$f(S) = \left( \bigcap_{i \in S} E_i \right) \bigcap \left( \bigcap_{i \in X \setminus S} \bar{E}_i \right), \ S \subseteq X$$

where $\bar{E}_i = V \setminus E_i$.

**Lemma 5.** $\{f(S) | S \in 2^X\}$ *is a partition of* $V$.

*Proof.* Clearly $f(S_i) \cap f(S_j) = \varnothing$ for all $S_i \neq S_j$, so elements in $\{f(S) | S \in 2^X\}$ are disjoint. Meanwhile, for every node $v \in V$, we can construct a $S = \{i | v \in E_i\}$ so that $v \in f(S)$. Therefore, the union of all elements in $\{f(S) | S \in 2^X\}$ spans $V$. □

Because Lemma 5 holds, for any $v \in V$ we can define the reverse function $f^{-1}(v) = S \Leftrightarrow v \in f(S)$. Here $f^{-1}$ is a signature function that represents a node in $V$ by a subset of $X$, whose physical meaning is the intersection of hyperedges in $\mathcal{H}$.

**Lemma 6.** *If $S_1 \cap S_2 \neq \varnothing, S_1, S_2 \subseteq X$, then for every $v_1 \in f(S_1)$ and $v_2 \in f(S_2)$, $(v_1, v_2)$ is an edge in H's projection G. Reversely, if $(v_1, v_2)$ is an edge in G, $f^{-1}(v_1) \cap f^{-1}(v_2) \neq \varnothing$.*

*Proof.* According to the definition of $f(S)$, $\forall i \in S_1 \cap S_2$, $v_1, v_2 \in E_i$. Appearing in the same hyperedge means that they are connected in $G$, so the first part is proved. If $(v_1, v_2)$ is an edge in $G$, there exists an $E_i$ that contains both nodes, so $f^{-1}(v_1) \cap f^{-1}(v_2) \supseteq \{i\} \neq \varnothing$. □

An *intersecting family* $\mathcal{F}$ is a set of non-empty sets with non-empty pairwise intersection, *i.e.* $S_i \cap S_j \neq \varnothing$, $\forall S_i, S_j \in \mathcal{F}$. Given a set $X$, a *maximal intersecting family of subsets*, is an intersecting family of set $\mathcal{F}$ that satisfies two additional conditions: (1) Each element of $\mathcal{F}$ is a subset of $X$; (2) No other subset of $X$ can be added to $\mathcal{F}$.

**Lemma 7.** *Given a a hypergraph $\mathcal{H} = (V, \{E_1, E_2, ...E_m\})$, its projection G, and $X = \{1, 2, ..., m\}$, the two statements below are true:*

- *If a node set $C \subseteq V$ is a maximal clique in G, then $\{f^{-1}(v)|v \in C\}$ is a maximal intersecting family of subsets of $X$.*

- *Reversely, if $\mathcal{F}$ is a maximal intersecting family of subsets of $X$, then $\cup_{S \in \mathcal{F}} f(S)$ is a maximal clique in G.*

*Proof.* For the first statement, clearly $\forall v \in V$, $f^{-1}(v) \subseteq X$. Because $C$ is a clique, every pair of nodes in $C$ is an edge in $G$. According to Lemma 6, $\forall v_1, v_2 \in C$, $f^{-1}(v_1) \cap f^{-1}(v_2) \neq \varnothing$. Finally, because $C$ is maximal, there does not exist a node $v \in V$ that can be added to $C$. Equivalently there does not exist a $S = f^{-1}(v)$ that can be added to $f^{-1}(C)$. Therefore, $f^{-1}(C)$ is maximal.

For the second statement, because $\mathcal{F}$ is an intersecting family, $\forall S_1, S_2 \subseteq \mathcal{F}$, $S_1 \cap S_2 \neq \varnothing$. According to Lemma 6, $\forall v_1, v_2 \in f(\mathcal{F})$, $(v_1, v_2)$ is an edge in $G$. Therefore, $f(\mathcal{F})$ is a clique. Also, no other node $v$ can be added to $f(\mathcal{F})$. Otherwise, $f^{-1}(v) \cup \mathcal{F}$ is still an intersecting family while $f^{-1}(v)$ is not in $\mathcal{F}$, which makes $\mathcal{F}$ strictly larger — a contradiction. Therefore, $\cup_{S \in \mathcal{F}} f(S)$ is a maximal clique. □

Lemma 7 shows there is a bijective mapping between a maximal clique and a maximal intersecting family of subsets. Given $\mathcal{H}$, $G$ and $X$, Counting the former is equivalent to counting the latter. The result is denoted as $\lambda(m)$ in the main text. Lemma 2.1 of Brouwer et al. (2013) gives an lower bound: $\lambda(m) \geq 2^{\binom{m-1}{\lceil m/2 \rceil - 1}}$.

### B.4 PROOF OF THEOREM 4

We start with some definitions. A *random finite set*, or RFS, is defined as a random variable whose value is a finite set. Given a RFS $A$, we use $\mathcal{S}(A)$ to denote $A$'s sample space; for a set $a \in \mathcal{S}(A)$ we use $P_A(a)$ to denote the probability that $A$ takes on value $a$. One way to generate a RFS is by defining the *set sampling* operator $\odot$ on two operands $r$ and $X$, where $r \in [0, 1]$ and $X$ is a finite set: $r \odot X$ is a RFS obtained by uniformly sampling elements from $X$ at sampling rate $r$, *i.e.* each element $x \in X$ has probability $r$ to be kept. Also, notice that the finite set $X$ itself can also be viewed as a RFS with only one possible value to be taken. Now, we generalize two operations, union and difference, to RFS as the following:

- **Union** $A \cup B$:

$$\mathcal{S}(A \cup B) = \{x|x = a \cup b, a \in A, b \in B\}$$
$$P_{A \cup B}(x) = \sum_{x = a \cup b, a \in A, b \in B} P_A(a) P_B(b)$$

- **Difference** $A \backslash B$:

$$\mathcal{S}(A \backslash B) = \{x|x = a \backslash b, a \in A, b \in B\}$$
$$P_{A \cup B}(x) = \sum_{x = a \backslash b, a \in A, b \in B} P_A(a) P_B(b)$$

With these ready, we have the following propositions that hold true for RFS $A$ and $B$:

(i) $\mathbb{E}(|A \cup B|) = \mathbb{E}(|B \cup A|)$

(ii) $\mathbb{E}(|A \cup B|) = \mathbb{E}(|A \backslash B|) + \mathbb{E}(|B|)$;

(iii) $\mathbb{E}(|A \cup B|) \geq \mathbb{E}(|A|)$, $\mathbb{E}(|A \cup B|) \geq \mathbb{E}(|B|)$;

(iv) $\mathbb{E}(|(r \odot X) \backslash Y|) = r|X \backslash Y| = \mathbb{E}(|X \backslash Y|)$;  ($X, Y$ are both set)

**Lemma 8.** *At iteration $(i + 1)$ when Algorithm 1 samples a cell $(n, k)$ (line 8), it reduces the gap between $q^*$ and the expected number of hyperedges it already collects, $q_i$, by a fraction of at least $\frac{r_{n,k}|Q_{n,k}|}{\beta}$:*

$$\frac{q^* - q_{i+1}}{q^* - q_i} \leq 1 - \frac{r_{i+1}|Q_{i+1}|}{\beta}$$

*Proof.*

$$q^* - q_i$$

$$= \mathbb{E}(|\cup_{j=1}^{z} r_j^* \odot \mathcal{E}_j|) - \mathbb{E}(|\cup_{j=1}^{i} r_j \odot \mathcal{E}_j|) \qquad (Thm.4\ setup)$$

$$\leq \mathbb{E}(|(\cup_{j=1}^{z} r_j^* \odot \mathcal{E}_j) \cup (\cup_{j=1}^{i} r_j \odot \mathcal{E}_j)]) - \mathbb{E}(|\cup_{j=1}^{i} r_j \odot \mathcal{E}_j|) \qquad (Prop.iii)$$

$$= \mathbb{E}(|(\cup_{j=1}^{z} r_j^* \odot \mathcal{E}_j) \backslash (\cup_{j=1}^{i} r_j \odot \mathcal{E}_j)|) \qquad (Prop.ii)$$

$$= \sum_{t=1}^{z} \mathbb{E}(|(r_t^* \odot \mathcal{E}_t) \backslash ((\cup_{j=1}^{t-1} r_t^* \odot \mathcal{E}_t) \cup (\cup_{j=1}^{i} r_j \odot \mathcal{E}_j))|) \qquad (Prop.ii)$$

$$\leq \sum_{t=1}^{z} \mathbb{E}(|(r_t^* \odot \mathcal{E}_t) \backslash (\cup_{j=1}^{i} r_j \odot \mathcal{E}_j)|) \qquad (Prop.iii)$$

$$= \sum_{t=1}^{z} r_t^* \mathbb{E}(|\mathcal{E}_t \backslash (\cup_{j=1}^{i} r_j \odot \mathcal{E}_j)|) \qquad (Prop.iv)$$

$$= \sum_{t=1}^{z} r_t^* |Q_t| \frac{\mathbb{E}(|\mathcal{E}_t \backslash (\cup_{j=1}^{i} r_j \odot \mathcal{E}_j)|)}{|Q_t|}$$

$$\leq \sum_{t=1}^{z} r_t^* |Q_t| \frac{\mathbb{E}(|\mathcal{E}_{i+1} \backslash (\cup_{j=1}^{i} r_j \odot \mathcal{E}_j)|)}{|Q_{i+1}|} \qquad (Alg.1,\ line\ 7)$$

$$= \beta \cdot \frac{r_{i+1} \mathbb{E}(|\mathcal{E}_{i+1} \backslash (\cup_{j=1}^{i} r_j \odot \mathcal{E}_j)|)}{r_{i+1}|Q_{i+1}|} \qquad (Def.\ of\ \beta)$$

$$= \frac{\beta}{r_{i+1}|Q_{i+1}|} \mathbb{E}(|(r_{i+1} \odot \mathcal{E}_{i+1}) \backslash (\cup_{j=1}^{i} r_j \odot \mathcal{E}_j)|) \qquad (Prop.iv)$$

$$= \frac{\beta}{r_{i+1}|Q_{i+1}|} (q_{i+1} - q_i) \qquad (Thm.4\ setup)$$

Therefore, $\frac{q^* - q_{i+1}}{q^* - q_i} \leq 1 - \frac{r_{i+1}|Q_{i+1}|}{\beta}$ $\qquad\square$

Now, according to our budget constraint we have

$$\sum_{n,k} (1 - \frac{r_{n,k}|Q_{n,k}|}{\beta}) = z - 1$$

$z$ is the total number of $(n, k)$ pairs where $r_{n,k} > 0$, which is a constant. Finally, we have

$$\frac{q^* - q}{q^*} = \prod_{i=0}^{z-1} \frac{q^* - q_{i+1}}{q^* - q_i} \leq \prod_{n,k} (1 - \frac{r_{n,k}|Q_{n,k}|}{\beta}) \leq (1 - \frac{1}{z})^z < \frac{1}{e}$$

Therefore $q > (1 - \frac{1}{e})q^*$.

## B.5 Errors I & II vs. "Type I & II" Errors

Note that here Errors I and II are different from the well-known Type I (false positive) and Type II (false negative) Error in statistics. In fact, Error I's are hyperedges that nest inside some other hyperedges, so they are indeed false negatives; Error II's can be either false positives or negatives. For example, in Fig.2 "Error II Pattern" : $\{v_1, v_3, v_5\}$ is a false positive, $\{v_1, v_5\}$ is a false negative.

# C  ADDITIONAL RELATED WORK

## C.1  COMPARISON BETWEEN GRAPHS AND HYPERGRAPHS

Previous works have compared the graph representations (*e.g.*, clique expansion) and hypergraph representations in various contexts, highlighting the concerning loss of higher-order information in graph representations. For example, Torres et al. (2021) provides a unified overview of various representations for complex systems: it notes the mathematical relationship between graphs and hypergraphs, warning against unthoughtful usage of graphs to model higher-order relationships. Similarly, Dong et al. (2020) also raises concerns about information loss when designing learning methods for hypergraphs based on their clique expansions.

Yoon et al. (2020) further empirically compares the performance of various methods in the hyper-edge prediction task when the methods are executed on different lower-order approximations of hypergraphs, including order-2 approximation (clique expansion), order-3 approximations (3-uniform projected hypergraphs), *etc.*; their experiments show that lower-order approximations especially struggle on more complex datasets or more challenging versions of tasks.

Our work significantly extends the notions in these previous works, formulating the central problem of interest (*i.e.* comparing graphs and hypergraphs, and mitigating the information loss) and giving it a systematical, rigorous study.

## C.2  RELATED METHODS TO HYPERGRAPH RECONSTRUCTION TASK

Besides the ones in Introduction, three lines of works are pertinent to the hypergraph reconstruction task discussed in this paper.

**Edge Clique Cover** is to find a minimal set of cliques that cover all the graph's edges. Erdös et al. (1966) proves that any graph can be covered by at most $[|V|^2/4]$ cliques. Conte et al. (2016) finds a fast heuristic for approximating the solution. Young et al. (2021) creates a probabilistic to solve the task. However, this line of work shares the "principle of parsimony", which is often impractical in real-world datasets.

**Hyperedge Prediction** is to identify missing hyperedges of an incomplete hypergraph from a pool of given candidates. Existing works focus on characterizing a node set's structural features. The methods span proximity measures Benson et al. (2018a), deep learning Li et al. (2020); Zhang et al. (2019), and matrix factorization Zhang et al. (2018). Despite the relevance, the task has a very different setting and focus from ours as mentioned in introduction.

**Community Detection** finds node clusters in which edge density is unusually high. Existing works roughly comes in two categories by the community types: disjoint Que et al. (2015); Traag et al. (2019), and overlapping Coscia et al. (2012); Palla et al. (2005). As mentioned, however, their focus on "relative density" is different from ours on cliques.

## C.3  COMPARING LEARNING-BASED HYPERGRAPH RECONSTRUCTION WITH HYPERGRAPH PREDICTION

As we observe in Table 3, the two hyperedge prediction methods have very poor performance. This is resulted from the incompatibility of the hypergraph prediction task with our task of hypergraph reconstruction, in particular:

- **Incompatibility of Input:** all hyperedge prediction methods, including the two baselines, require a (at least partially observed) hypergraph as input, and they must run on hyperedges. In hypergraph reconstruction task, we don't have this as input. We only have a projected graph.

- **Incompatibility of output:** hyperedge prediction methods can only tell us whether a potential hyperedge can really exist. They cannot tell where a hyperedge is, given only a projected graph as input. In other words, they only do classification, rather than generation. This is also the key issue that our work has gone great lengths to address.

In order to run hyperedge prediction methods on hypergraph reconstruction, we have to make significant adaptations. First, we must treat all edges in the input projected graph as existing hyperedges. Second, it is only fair that we sample cliques from projected graph as "potential hyperedge" for classification completely at random, until we reach our computing capacity. This is because none of the hyperedge prediction methods mentions or concerns about this procedure. These two adaptations enable hyperedge prediction methods to run on hypergraph reconstruction tasks, but at the cost of vastly degraded performance.

## D    MORE DISCUSSIONS ON THE SUPERVISED HYPERGRAPH RECONSTRUCTION APPROACH

### D.1    COMPLEXITY OF $\rho(n, k)$

The main complexity of $\rho(n, k)$ involves two parts: **(a)** $\mathcal{M}$; **(b)** $\mathcal{E}_{n,k}$ for all valid $(n, k)$.

**(a)**'s complexity is $O(|\mathcal{M}|)$ as mentioned in Sec.2. Though in worst case $|\mathcal{M}|$ can be exponential to $|V|$, in practice we often observe $|\mathcal{M}|$ on the same magnitude order as $|\mathcal{E}|$ (see Table 1), which is an interesting phenomenon. Please see our discussion below for more details.

**(b)** requires matching size-$n$ maximal cliques with size-$k$ hyperedges for each valid $(n, k)$ pair. The key is that in real-world data, the average number of hyperedges incident to a node is usually a constant independent from the growth of $|V|$ or $|\mathcal{M}|$ (see also $\bar{d}(V)$ in Table 2), known as the sparsity of hypergraphs Kook et al. (2020). This property greatly reduces the average size of search space for all size-$k$ hyperedges in a size-$n$ maximal clique from $|\mathcal{E}|$ to $n\bar{d}(V)$. As we see both $n$ and $\bar{d}(V)$ are typically under 50 in practice. **b**'s complexity can still be viewed as $O(|\mathcal{M}|)$. Therefore, the total complexity for computing $\rho(n, k)$ is $O(|\mathcal{M}|)$. Sec.5.2 provides more empirical evidence.

**More on the complexity of extracting maximal cliques:**
As discussed above, the complexity of extracting data inputs for Algorithm1 is bounded by the number of maximal cliques in the projected graph, and for thinking about this, we believe it's useful to distinguish between the worst case and the cases we encounter in practice. For the worst case, the number of maximal cliques indeed can be (super-)exponential to the number of hyperedges, as our Theorem 3 proved. We agree with the reviewer on this point.

The interesting observation here is that in practice we don't typically witness such an explosion when we try to enumerate all maximal cliques in various datasets, see Table 5 in Appendix for example. This contrast between the worst case and the cases encountered in practice is of course a common theme in network analysis, where research has often tried to provide theoretical or heuristic reasons why the worst-case behavior of certain methods doesn't seem to generally occur on the kinds of real-world network that arise in practice. This theme has been explored in a number of lines of work for problems that require the enumeration of maximal cliques.

In particular, there are multiple lines of work that try to give theoretical explanations for why real-world graphs generally have a tractable number of maximal cliques, thereby making algorithms that use maximal clique enumeration feasible. The reviewer's point is correct that some of these are related to the sparsity of the input graph, but they also include other structural features typically exhibited by real-world network structures. Here we include brief discussions on two metrics relevant to this:

- $k$**-degeneracy.** The degeneracy of an n-vertex graph G is the smallest number $k$ such that every subgraph of $G$ contains a vertex of degree at most $k$. The $k$ here is often used as a measure of how sparse a graph is, in a way that captures subtler structural information than simply the average degree, and with a smaller $k$ indicating a sparser graph. Eppstein et al. (2010) found that a graph of n nodes and degeneracy $k$ can have at most $(n - k)3^{k/3}$ maximal cliques. This result explains as it restricts the size of hyperedges.

- $c$**-closure.** An undirected graph $G$ is $c$-closed if, whenever two distinct vertices $u, v$ have at least $c$ common neighbors, $(u, v)$ is an edge of G. The number c here measures the strength of triadic closure of this graph. Fox et al. (2020) shows that any (weakly) $c$-closed graph on n vertices has at most $3^{(c-1)/3}n^2$ maximal cliques. Our paper is benefiting from the tractable

number of maximal cliques on real-world networks, on which there's been a lot of progress in the theoretical underpinnings via these and follow-up papers.

## D.2 NUMERICAL STABILITY OF $\rho(n, k)$

A desirable property of $\rho(n, k)$ to be a hypergraph statistic is its robustness to small distribution shifts of the hypergraphs. Here we report a study of $\rho(n, k)$'s numerical stability based on simulation.

First, we introduce a simple generative model for hypergraphs:

$$\mathcal{H} \sim P(n, k, \mathbf{m})$$

where $n$ is the total number of nodes, $k$ is the size of the largest hyperedge (measured by number of nodes in that hyperedge), and $\mathbf{m}$ is a vector of length $(k - 1)$ whose $i$-th element denotes the number of hyperedges of size $(i + 1)$. This model generates a random hypergraph by sampling from $n$ nodes $\mathbf{m}[1]$ hyperedges of size 2, $\mathbf{m}[2]$ hyperedges of size 3, ... $\mathbf{m}[k - 1]$ hyperedges of size k.

Next, we study how a small perturbation to vector $\mathbf{m}$ results in the faction of change in the distribution of $\rho(n, k)$'s. Fixing $n$ and $k$, for each $\mathbf{m}$ we add a random noise (*i.e.* a vector $\Delta\mathbf{m}$) at the strength of 5%, *i.e.* $|\Delta\mathbf{m}|/|\mathbf{m}| = 0.05$. As a result of the added noise, $\rho(n, k)$ would become $\hat{\rho}(n, k)$ We can therefore quantify the instability of $\rho(n, k)$ with respect to $\mathbf{m}$ as

$$\text{instability} = \frac{\sum_{n,k}(\rho(n, k) - \hat{\rho}(n, k))^2 / \sum_{n,k}(\rho(n, k))^2}{|\Delta\mathbf{m}|/|\mathbf{m}|}$$

In our experiment, for each $\mathbf{m}$ we repeat the measurement for this stability quantifier for 10 times. We study a simple case where $k = 5$, $n$ ranges from 10 to 60. In order to mimic the sparsity of the hypergraphs in real world, we further set each element in $\mathbf{m}$ to be $n$ (so that the hypergraph's density is on the order of O(1)). Here is the result of the instability test:

| n | 10 | 20 | 30 | 40 | 50 | 60 | 70 |
|---|---|---|---|---|---|---|---|
| **instability** | 0.45 | 0.58 | 0.47 | 0.39 | 0.29 | 0.20 | 0.18 |

Table 5: Numerical instability of $\rho(n, k)$ with regard to small perturbations in the hypergraph.

We observe that the instabilities are smaller that 1. This means that the relative change in the distribution of rho(n,k) is actually smaller than the relative change in the distribution of hyperedges captured by hyperedge numbers m. And as n goes large the influence of the disturbance decades. Finally, we also acknowledge that the result of this simulation may not apply universally, and that a fully theoretical analysis of the numerical stability of rho(n,k) is unavailable to us at this point.

## D.3 MORE DISCUSSION ON ALGORITHM 1

**Relating to Errors I & II.** The effectiveness of the clique sampler can also be interpreted by the reduction of Errors I and II. Taking Fig.4a as an example: by learning which non-diagonal cells to sample, the clique sampler essentially reduces Error I as well as the false negative part of Error II; by learning which diagonal cells to sample, it further reduces the false positive part of Error II.

**Relating to Standard Submodular Optimization.** There are two distinctions between our clique sampler and the standard greedy algorithm for submodular optimization.

- The standard greedy algorithm runs deterministically on a set function whose form is already known. In comparison, our clique sampler runs on a function defined over Random Finite Sets (RFS) whose form can only be statistically estimated from the data.
- The standard submodular optimization problem forbids picking a set fractionally. Our problem allows fractional sampling from an RFS (*i.e.* $r_{n,k} \in [0, 1]$).

We can see from the Proof of Theorem 4 above that it is harder to prove the optimality of our clique sampler than to prove for the greedy algorithm for Standard Submodular Optimization.

**Precision-Recall Tradeoff.** For each dataset, $\beta$ should be specified manually. What's the best $\beta$? Clearly a larger $\beta$ yields a larger $q$, thus a higher recall $\frac{q}{|\mathcal{E}|}$ in samples. On the other hand, a larger

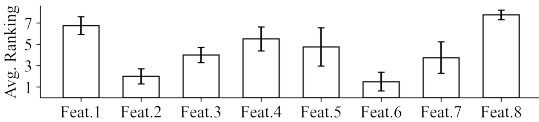

Figure 7: Average rankings of the 8 count features. More imporant features have smaller ranking numbers.

$\beta$ also yields a lower precision $\frac{q}{\beta}$, as sparser regions get sampled. $\frac{q}{\beta}$ being too low harms sampling quality and later the training. Such tradeoff necessitates more calibration of $\beta$. We empirically found it often good to search $\beta$ in a range that makes $\frac{q}{|\mathcal{E}|} \in [0.6, 0.95]$, with more tuning details in Appendix.

**Complexity.** The bottleneck of Algo. 1 is **UPDATE**. In each iteration after a $k$ is picked, **UPDATE** recomputes $(|\Gamma_k \cup \mathcal{E}_{n,k}| - |\Gamma_k|)$ for all $n \in \omega_k$, which is $O(\frac{|\mathcal{E}|}{N})$. Empirically we found the number of iterations under the best $\beta$ always $O(N)$. $N$ is the size of the *maximum* clique, and mostly falls in $[10, 40]$ (see Fig.4b). Therefore, on expectation we would traverse $O(N)O(\frac{|\mathcal{E}|}{N}) = O(|\mathcal{E}|)$ hyperedges if $|\mathcal{E}_{n,k}|$ distributes evenly among different $k$'s. In the worst case where $|\mathcal{E}_{n,k}|$'s are deadly skewed, this degenerates to $O(N|\mathcal{E}|)$.

### D.4 COUNT FEATURES

We define a target clique $C = \{v_1, v_2, ..., v_{|C|}\}$. The 8 features are:

1. size of the clique: $|C|$;

2. avg. node degree: $\frac{1}{|C|} \sum_{v \in C} d(v)$;

3. avg. node degree (recursive): $\frac{1}{|C|} \sum_{v \in C} \frac{1}{|\mathcal{N}(v)|} \sum_{v' \in \mathcal{N}(v)} d(v')$;

4. avg. node degree *w.r.t.* max cliques: $\frac{1}{|C|} \sum_{v \in C} |\{M \in \mathcal{M} | v \in M\}|$;

5. avg. edge degree *w.r.t.* max cliques: $\frac{1}{|C|} \sum_{v_1, v_2 \in C} |\{M \in \mathcal{M} | v_1, v_2 \in M\}|$;

6. binarized "edge degree" (*w.r.t.* max cliques): $\prod_{v_1, v_2 \in C} \mathbb{1}_{[e]}$, where $e = \sum_{v_1, v_2 \in C} |\{M \in \mathcal{M} | v_1, v_2 \in M\}| > 1$;

7. avg. clustering coefficient: $\frac{1}{|C|} \sum_{v \in C} cc(v)$, where $cc$ is the clustering coefficient of node $v$ in the projection;

8. avg. size of encompassing maximal cliques: $\frac{1}{|\mathcal{M}^C|} \sum_{M \in \mathcal{M}^C} |M|$, where $\mathcal{M}^C = \{M \in \mathcal{M} | C \subseteq M\}$;

Notice that avg. clustering coefficient is essentially a normalized count of the edges between direct neighbors. All the above features, except features 2,3,7, are new contributions of our paper. Features 2,3,7 are very commonly used count features.

**Feature Rankings**. We study the relative importance of the 8 features by an ablation study. For each dataset, we ablate the 8 features one at a time, record the performance drops, and use those values to rank the 8 features. We repeat this for all datasets, obtaining the 8 features' average rankings, shown in Fig.7. More imporant features have smaller ranking numbers. Interestingly we found that the most important feature has a very concrete physical meaning: it indicates whether each edge of the clique has existed in at least two maximal cliques of the projected graph. Remarkably, this is very similar to the simplicial closure phenomenon we've seen in in Benson et al. (2018a).

## E BROADER IMPACTS

Extra care should be taken when the hypergraph to be reconstructed involves human subjects. For example, in Fig.8a, the reconstruction of DBLP coauthorship hypergraph has higher accuracy on larger hyperedges, missing more hyperedges of sizes 1 and 2. In other words, papers with fewer authors seem to be harder to recover in this case. The technical challenge in solving this problem resides in the Error I patterns as discussed in the main text. Meanwhile, small hyperedges also contain less structural information to be used by the classifier. In the broader sense, this issue could lead to concerns that marginalized groups of people are not given equal amount of attention as other groups.

| Dataset | $|V|$ | $|\mathcal{E}|$ | $\mu(\mathcal{E})$ | $\sigma(\mathcal{E})$ | $\bar{d}(V)$ | $|\mathcal{M}|$ |
|---|---|---|---|---|---|---|
| Enron Benson et al. (2018a) | 142 recipients | 756 emails | 3.0 | 2.0 | 16 | 362 |
| DBLP Benson et al. (2018a) | 319,916 authors | 197,067 papers | 3.0 | 1.7 | 1.8 | 166,571 |
| P. School Benson et al. (2018a) | 242 students | 6,352 chats | 2.4 | 0.6 | 64 | 15,017 |
| H. School Benson et al. (2018a) | 327 students | 3,909 chats | 2.3 | 0.5 | 28 | 3,279 |
| Foursquare Young et al. (2021) | 2,334 eateries | 1,019 footprints | 6.4 | 6.5 | 2.8 | 8,135 |
| Hosts-Virus Young et al. (2021) | 466 hosts | 218 virus | 5.6 | 9.0 | 2.6 | 361 |
| Directors Young et al. (2021) | 522 directors | 102 boards | 5.4 | 2.2 | 1.2 | 102 |
| Crimes Young et al. (2021) | 510 victims | 256 homicides | 3.0 | 2.3 | 1.5 | 207 |

Table 6: $\mathcal{E}$ is the set of hyperedges; $\mathcal{E}'$ is the set of hyperedges not nested in any other hyperedges; $\mathcal{M}$ is the set of maximal cliques in $G$. **Error I, II** result from the violation of conformal and Sperner properties, respectively. Error I $= \frac{|\mathcal{E}\setminus\mathcal{E}'|}{|\mathcal{E}\cup\mathcal{M}|}$, Error II $= \frac{|\mathcal{M}\setminus\mathcal{E}'|+|\mathcal{E}'\setminus\mathcal{M}|}{|\mathcal{E}\cup\mathcal{M}|}$.

To mitigate the risk of this issue, we propose improvement to three places in the reconstruction pipeline, and encourage follow-up research into those directions. First, the training hypergraph $\mathcal{H}_0$ can be improved towards more emphasis on smaller hyperedges. In practice, for example, this can be achieved by collecting more data instances from marginalized social groups. Second, we can also adjust the clique sampler so that it allocates more sampling budget to small hyperedges. Technically speaking, we can manually assign larger values to the $r_{n,1}$'s and the $r_{n,2}$'s, which originally are parameters to be learned. Third, the hyperedge classifier may also be improved towards better characterization of small hyperedges. One promising direction to achieve this is to utilize node or edge attributes, which, similar to the first measure, also boils down to more data collected on marginalized groups.

# F EXPERIMENTS

## F.1 EXPERIMENTAL SETUP - ADDITIONAL DETAILS

**Selection Criteria and Adaptation of Baselines**. For *community detection*, the criteria are: 1. community number must be automatically found; 2. the output is overlapping communities. Based on them, we choose the most representative two. We tested Demon and found it always work best with min community size = 1 and $\epsilon = 1$. To adapt CFinder we search the best $k$ between $[0.1, 0.5]$ quantile of hyperedge sizes on $\mathcal{H}_0$. For *hyperedge prediction*, we ask that they cannot rely on hypergraphs for prediction, and can only use the projection. Based on that we use the two recent SOTAs, Zhang et al. (2018; 2019). We use their default hyperparameters for training. For Bayesian-MDL we use its official library in graph-tools with default hyperparameters. We implemented the best heuristic in Conte et al. (2016) for clique covering. Both our method and Bayesian-MDL use the same maximal clique algorithm (*i.e.* the Max Clique baseline) as a preprocessing step.

**Datasets**. The first 4 datasets in Table 2 are from Benson et al. (2018a); the rest are from Young et al. (2021). All source links and data can be found in submitted code.

**Generating training & query set**. To generate a training set and a query set, we follow two common standards to split the collection of hyperedges in each dataset: (1) For datasets that come in natural segments, such as DBLP and Enron whose hyperedges are timestamped, we follow their segments so that training and query contain two disjoint and roughly equal-sized sets of hyperedges. For DBLP, we construct $\mathcal{H}_0$ from year 2011 and $\mathcal{H}_1$ from year 2010; for Enron, we use 02/27/2001, 23:59 as a median timestamp to split all emails into $\mathcal{H}_0$ (first half) and $\mathcal{H}_1$ (second half). (2) For all the other datasets that lack natural segments, we randomly split the set of hyperedges into halves.

We note that the first standard has less restrictions on the data generation process than the second standard, which follows a more ideal setting and is most widely seen as the standard train-test split in ML evaluations. We therefore also introduce the settings of semi-supervised learning and transfer learning in Sec.5.3 to compensate. We consider these settings together as a relatively comprehensive suite for validating both the proposed learning-based reconstruction problem and its solution pipeline.

To enforce inductiveness, we also randomly re-index node IDs in each split. Finally, we project $\mathcal{H}_0$ and $\mathcal{H}_1$ to get $G_0$ and $G_1$ respectively.

**Hyperparameter Tuning.** For Demon, we tested all combinations of its two hyperparameters (min_com_size, epsilon), and found that on all datasets the best combination is (1, 1). For CFinder, we tuned its hyperparameter k by search through $10\%, 20\%, \ldots, 50\%$ quantiles of distribution of hyperedge sizes in the dataset. For CMM, we set the number of latent factors to 30. For Hyper-SAGNN, we set the representation size to 64, window size to 10, walk length to 40, the number of walks per vertex to 10. We compare the two variants of Hyper-SAGNN: the encoder variant and the random walk variant, and chose the latter which consistently yields better performance. The baselines of Bayesian-MDL, Maximal Cliques and Clique Covering do not have hyperparameters to tune.

**Training Configuration.** For models requiring back propagation, we use cross entropy loss and optimize using Adam for 2000 epochs and learning rate 0.0001. Those with randomized modules are repeated 10 times with different seeds.Regarding the tuning of $\beta$, we found the best $\beta$ by training our model on $90\%$ training data and evaluated on the rest $10\%$ training data. The best values are reported in our code instructions.

**Machine Specs.** All experiments including model training are run on Intel Xeon Gold 6254 CPU @ 3.15GHz with 1.6TB Memory.

### F.2  ANALYSIS OF RECONSTRUCTIONS

#### F.2.1  BASIC PROPERTIES

We characterize the topological properties of the reconstruction using the middle four columns of Table 2: $[|\mathcal{E}|, \mu(\mathcal{E}), \sigma(\mathcal{E}), \bar{d}(V)]$. $|V|$ is excluded as it is known from the input. For each (dataset, method) combination, we analyze the reconstruction and obtain a unique property vector. We use PCA to project all (normalized) property vectors into 2D space, visualized in Fig.8a.

In Fig.8a, colors encode datasets, and marker styles encode methods. Compared with baselines, SHyRe variants ( $\bigcirc$ and $\times$) produce reconstructions more similar to the ground truth ($\blacksquare$). The reasons are two-fold: (1) SHyRe variants have better accuracy, which encourages a more aligned property space; (2) This is a bonus of our greedy Algo. 1, which tends to pick a cell from a different column in each iteration. Cells in the same column has diminishing returns due to overlapping of $\mathcal{E}_{n,k}$ with same $k$, whereas cells in different columns remain unaffected as they have hyperedges of different sizes. The inclination of having diverse hyperedge sizes reduces the chance of a skewed distribution.

Markers of the same colors are cluttered, meaning that most baselines work to some extent despite low accuracy sometimes. Fig.8a also embeds a histogram for the size distribution of the reconstructed hyperedges on DBLP. SHyRe's distribution aligns decently with the ground truth, especially on large hyperedges. Some errors are made on sizes 1 and 2, which are mostly the nested cases in Fig.2.

Fig.8b visualizes a portion of the actual reconstructed hypergraph by SHyRe-count on DBLP dataset. The caption includes more explanation and analysis.

#### F.2.2  ADVANCED PROPERTIES

Inspired by Lee et al. (2022), we further compare some of the advanced structural properties between the original hypergraphs and the reconstructed hypergraphs. The advanced structural properties include simplicial closure (Benson et al., 2018a), degree distribution, singular-value distribution, density, and diameter.

For simplicial closure, density, and diameter, we quantify the alignment by: $|x_1 - x_2|/\max(x_1, x_2)$, so that it is a continuous value in $[0, 1]$; $x_1, x_2$ are the values of ground truths and reconstructions, respectively. This gives us a unified measurement that can be compared across different datasets. A **larger** value here indicates better alignment. For the alignment degree and singular values, we treat each of them as a probability density function and report the cross entropy between the two distributions. A **smaller** value indicates better alignment.

The result is presented in Table 7. We observe that our method works well at recovering the density and diameter. We also find that the degree of alignment on simplicial closure between ground truth and reconstruction seems to be correlated well with the Jaccard scores reported in our main Table 3. It is not easy though to obtain an intuitive sense of the alignment between two distributions measured by

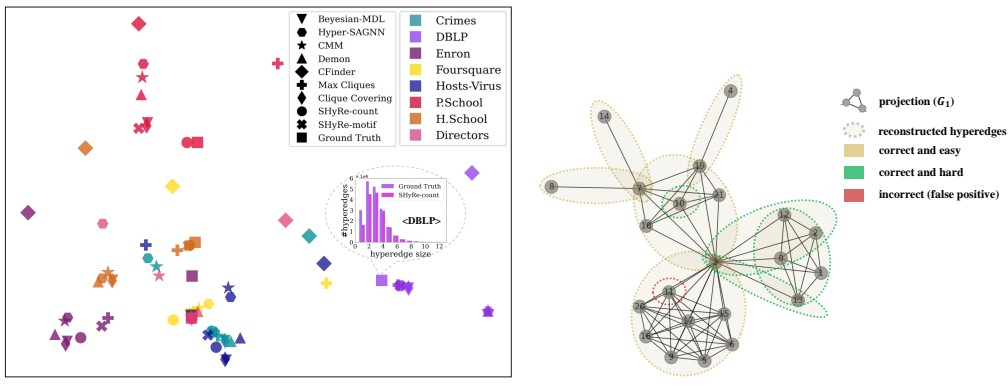

(a)  (b)

Figure 8: **(a)** 2D embeddings of statistical properties of reconstructed hypergraphs. Colors encode datasets; markers encode methods. SHyRe produces reconstructions closest to ground truth (■). **(b)** A part of SHyRe-motif's reconstructed hypergraph on DBLP dataset. The black edges are $G_1$. The shaded ellipsis are the reconstructed hyperedges. Those with green dashed edges are difficult to reconstruct if not using training data.

cross entropy. However, the ordering of the cross entropy on all datasets seems to decently correlate with the ordering of the Jaccard scores.

| | DBLP | Enron | P.School | H.School | Foursquare | Hosts-Virus | Directors | Crimes |
|---|---|---|---|---|---|---|---|---|
| Simplicial Closure | 0.89 | 0.29 | 0.38 | 0.46 | 0.74 | 0.49 | 1.00 | 0.78 |
| Density ($|\mathcal{E}|/|\mathcal{V}|$) | 0.92 | 0.95 | 0.82 | 0.95 | 0.90 | 0.87 | 1.00 | 0.86 |
| Diameter | 1.00 | 0.83 | 1.00 | 1.00 | 1.00 | 1.00 | 1.00 | 1.00 |
| Degree Distribution | 6.69e-7 | 9.78e-2 | 3.46e-2 | 1.37e-2 | 1.91e-3 | 1.10e-2 | 0 | 8.9e-6 |
| Singular Value Distribution | 1.73e-10 | 3.83e-4 | 5.61e-5 | 8.49e-4 | 4.83e-6 | 6.40e-1 | 0 | 1.29e-5 |

Table 7: Comparative analysis of advanced properties of the ground-truth hypergraphs and reconstructed hypergraphs. For simplicial closure, density, and diameter, we quantify the alignment by: $|x_1 - x_2|/\max(x_1, x_2)$, so that it is a continuous value in $[0, 1]$; $x_1, x_2$ are the values of ground truths and reconstructions, respectively. A **larger** value indicates better alignment. For the alignment degree and singular values, we treat each of them as a probability density function and report the cross entropy between the two distributions. A **smaller** value indicates better alignment.

### F.3 CORRELATION ANALYSIS BETWEEN HYPERGRAPH PROPERTIES AND RECONSTRUCTION PERFORMANCE

On some datasets, our proposed method significantly outperforms baselines; on some other datasets, our improvement is less prominent. To understand what properties of the dataset makes our proposed method most suitable, we conduct a mini-study to interpret Table 3.

In this mini-study, we fit a linear regression model to the performance gap ($y$) between our method and the best baseline, using the four basic properties of hypergraphs ($X$) listed in Table 6

- average hyperedge size ($\mu(\mathcal{E})$);
- standard deviation of hyperedge size ($\sigma(\mathcal{E})$);
- node degree in hypergraph ($d(\bar{V}) = |\mathcal{E}|/|V|$);
- number of maximal cliques in projection ($|\mathcal{M}|$);

The performance gap is computed by:

$$\text{best performance of SHyRe variants} - \text{best performance of baselines} \quad (1)$$

based on Table 3. For example, for DBLP the performance gap is $81.19 - 73.08 = 8.11$.

The resultant four coefficients are shown in Figure 9. The bar plot shows that:

- Our proposed method has more advantage on datasets with large average node degree, such as Enron, P.School, and H.School. Notice that large average node degree means that the

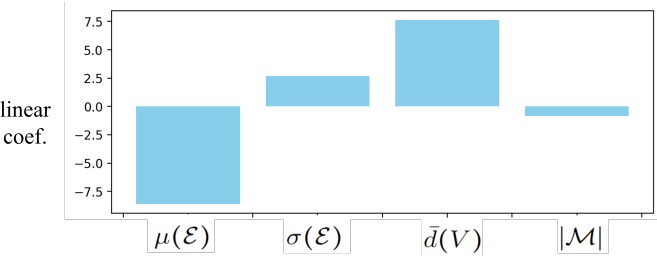

Figure 9: Contributions of different dataset properties to the performance gap between our method and baselines. We fit a linear regression model to the performance gap using the four properties shown in the plot.

hypergraph has densely overlapping hyperedges. For example, in P.School dataset, each node appears in an average of 64 hyperedges. The dense overlapping of hyperedges, as we have analyzed in Section 3,create highly challenging cases for reconstruction. This is why our proposed method stands out,as it can effectively leverages information embedded in the training hypergraphs.

- Our proposed method has generally less advantage on datasets with large average hyperedge size, such as Foursquare and Directors. A possible explanation is that, compared with small hyperedge, projections from large hyperedges are less likely to mix up with each other, making the reconstruction easier. Meanwhile, baselines like max clique, clique covering, and Bayesian-MDL, all favor large cliques in reconstruction by their design. Therefore, we observe more competitive baselines on datasets with large hyperedges.

### F.4    USE CASE 1: NODE RANKING IN PROTEIN-PROTEIN INTERACTION (PPI) NETWORKS

**Background.** We applied our method to PPI networks, recovering multiprotein complexes from pairwise protein interaction data. More than half of the proteins in nature form multiprotein complexes, which perform many fundamental functions such as DNA transcription, energy supply, *etc.* (Spirin & Mirny, 2003). Mainstream laboratory-based methods for directly detecting proteins in multiprotein complexes, such as TAP-MS (Rigaut et al., 1999) or LUMIER (Blasche & Koegl, 2013), are known to be expensive and error-prone (Brückner et al., 2009; Hoch & Soriano, 2015). Alternatively, well-studied high-throughput methods such as Yeast 2-hybrid screening (Y2H) (Young, 1998) captures pairwise protein interactions. Brückner et al. (2009) mentions that "MS is less accessible than Y2H due to the expensive large equipment needed. Thus, a large amount of the data so far generated from protein interaction studies have come from Y2H screening. " However, many studies (Klimm et al., 2021; Spirin & Mirny, 2003; Ramadan et al., 2004) found that the pairwise PPI network produced by Y2H obscures much information compared to the hypergraph modeling of multiprotein complexes. This is a natural setting for the hypergraph reconstruction task.

Our experiment follows the transfer learning's setting. We seek to recover multiprotein complexes from a dataset of pairwise protein interactions, Reactome Croft et al. (2010), given access to a different but similar dataset of multiprotein complexes, hu.MAP 2.0 Drew et al. (2017). After preprocessing, the training dataset has 6,292 nodes (proteins), 5119 ground truth hyperedges (multiprotein complexes); the query dataset has 8,243 nodes, 6,688 hyperedges. Our SHyRe-motif algorithm achieves a Jaccard accuracy of 0.43 (precision 0.89, recall 0.45). This means that we sucessfully recover around half of all multiproteins complexes in the query dataset, and around 90% of those recovered are correct.

**Node degrees for ranking protein essentiality.** Many studies Xiao et al. (2015); Estrada (2006); Klimm et al. (2021) found that the number of interactions that one protein participates in highly correlate to the protein's essentiality in the system. The best measure for this is hypergraph node degrees. In practice, node degrees in pairwise PPI networks are often used instead — a compromise due to the technical constraints mentioned.

Ranking the proteins with highest node degrees produces Table 8, the top-10 lists for their corresponding upstream gene names in the original multiproteins hypergraph ($\mathcal{H}$), the projected pairwise PPI network($G$), and the recovered multiprotein hypergraph ($\tilde{\mathcal{H}}$). The recovered hypergraph produces a list much closer to the ground truth than the pairwise graph does. Also, notice the different positions

of UBB and GRB2 in all lists: GRB2 encodes the protein that binds the epidermal (skin) growth factor receptor; UBB is one of the most conserved proteins, playing major roles in many critical functions including abnormal protein degradation, gene expression, and maintenance of chromatin structure. Therefore, UBB is arguably more essential than GRB2 in cellular functions, despite interacting with fewer proteins in total. The middle list produced by our algorithm precisely captures this subtlety.

We further use node degrees in the ground truth hypergraph as a reference, and compare their correlation with node degrees in the projected graph and the recovered hypergraph, respectively. The latter was found to have a Pearson correlation of 0.93, higher than the former of 0.88 ($p < 0.01$). This means that our method helps recover protein essentiality information that is closer to the ground truth.

| Ground Truth ($\mathcal{H}$) | UBB | UBC | RPS27A | UBA52 | GRB2 | JAK2 | JAK1 | SUMO1 | FGF2 | FGF1 |
|---|---|---|---|---|---|---|---|---|---|---|
| **Recovered ($\hat{\mathcal{H}}$)** | UBB | UBC | RPS27A | UBA52 | GRB2 | SUMO1 | ITGB1 | JAK2 | JAK1 | PIK3R1 |
| **Pairwise PPI ($G$)** | GRB2 | UBB | UBC | RPS27A | UBA52 | FGF2 | FGF1 | FGF9 | FGF17 | FGF8 |

Table 8: Top-10 most essential genes ranked by node degrees in different formalisms of hypergraphs/graph. Note that the recovered hypergraph yields a list much closer to the ground truth than the pairwise PPI does.

### F.5 USE CASE 2: LINK PREDICTION

As the second use case, we conduct a mini-study to compare the performance of link prediction (as a downstream task) on the projected graph $G$, the reconstructed hypergraph $\hat{\mathcal{H}}$, and the original hypergraph $\mathcal{H}$. To ensure fair comparison, all results are obtained using the same base model of a two-layered Graph Convolutional Network (GCN), including those on hypergraphs: The initial node features are initialized using one-hot encodings; to utilize hypergraph features when training on $\hat{\mathcal{H}}$ and $\mathcal{H}$, we append several structural features of hyperedges to the final link embeddings, including: the number of common neighbors of the two end nodes, the number of hyperedges associated with each of the two end nodes and the average size of the hyperedges associated with each of the two end nodes, and the shortest distance (minimum number of hyperedges to traverse) between the two end nodes.

The results are shown in Table 9. We observe that the numerical results show that the reconstructed hypergraphs are indeed helpful to link prediction task compared to the projection graph: 3 out 4 datasets if measured by AUC, and 4 out of 4 datasets if measured by Recall. Cross referencing our Table 3, we can also see the general trend that a better-reconstructed hypergraph would lead to more boost in link prediction performance. We also note that these results are not to demonstrate hypergraph reconstruction as a state-of-the-art method for link prediction. It, however, provides a piece of evidence that the reconstructed hypergraphs are a good form of intermediate representation which, compared to their projected graphs, is more informative and helpful in downstream tasks.

| | AUC-$G$ | AUC-$\hat{\mathcal{H}}$ | AUC-$\mathcal{H}$ | Recall-$G$ | Recall-$\hat{\mathcal{H}}$ | Recall-$\mathcal{H}$ |
|---|---|---|---|---|---|---|
| DBLP | $92.82 \pm 0.11$ | $93.54 \pm 0.10$ | $93.69 \pm 0.11$ | $67.82 \pm 0.39$ | $69.12 \pm 0.38$ | $71.44 \pm 0.28$ |
| Enron | $84.45 \pm 0.22$ | $84.66 \pm 0.23$ | $86.51 \pm 0.16$ | $68.10 \pm 0.50$ | $68.62 \pm 0.47$ | $70.59 \pm 0.42$ |
| Foursquare | $86.56 \pm 0.24$ | $87.71 \pm 0.24$ | $87.77 \pm 0.24$ | $75.04 \pm 0.38$ | $75.64 \pm 0.38$ | $75.01 \pm 0.35$ |
| Directors | $85.02 \pm 0.30$ | $86.33 \pm 0.49$ | $86.48 \pm 0.45$ | $83.31 \pm 0.21$ | $84.00 \pm 0.25$ | $83.31 \pm 0.22$ |

Table 9: Comparing the performance of link prediction (as a downstream task) on the projected graph $G$, the reconstructed hypergraph $\hat{\mathcal{H}}$, and the original hypergraph $\mathcal{H}$.

Again, we want to emphasize that the goal of this mini-study here **isn't to demonstrate the superiority of learning-based hypergraph reconstruction against the current state-of-the-art method for link prediction**. In fact, SOTA methods for link prediction is often end-to-end customized, in which case the reconstructed hypergraph is not necessary. We reconstruct hypergraphs, instead of end-to-end training a model for each downstream task with reconstructing hypergraphs, because we do not know the exact downstream task when we do reconstruction, and we may not even be the person to do the downstream task. Reconstructed hypergraphs can be viewed as an intermediate product for more general purpose and versatile usage, which is similar to the role word embeddings play in language tasks.

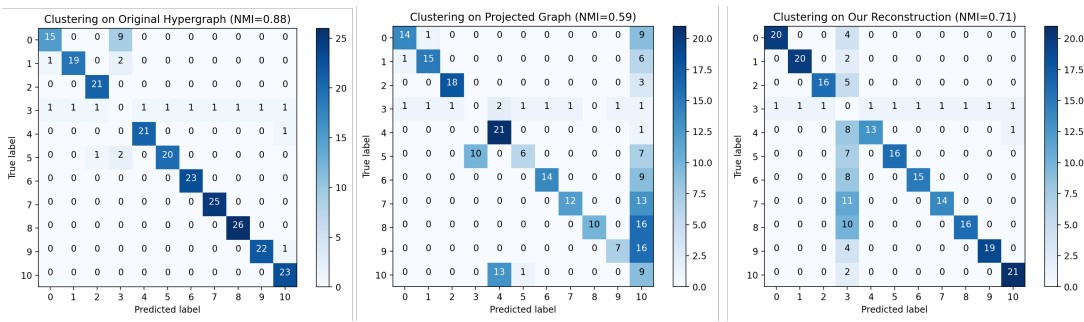

Figure 10: Performance (NMI) and confusion matrix of spectral clustering on the original hypergraph, projected graph, and our reconstructed hypergraph.

## F.6 USE CASE III: NODE CLUSTERING

In this third use case, we demonstrate the benefits of reconstructed hypergraphs via another classical task: node clustering. The dataset we use is P.School, which is also used in our main experiment. Its statistics can be found in Table 6. In this dataset, each node represents a student, and each hyperedge represents a face-to-face interaction among multiple students, recorded by their body-worn sensors during a period. Meanwhile, each node is labeled as one of the 11 classrooms to which the student belongs. We treat these node labels as the ground truth for clustering.

Similar to the first two use cases, we conduct clustering on the original hypergraph, the projected graph, and the reconstructed hypergraph. To ensure fairness, we stick to spectral clustering as the clustering method. Notice that spectral clustering can be easily generalized from graphs to hypergraphs, and has been used as a classical clustering method for hypergraphs for a long time (Zhou et al., 2006). We use normalized mutual information (NMI) and confusion matrix to evaluate the performance. A larger NMI indicates better performance.

Figure 10 shows the result. We see that our reconstructed hypergraph has much better NMI than the projected graph does. We can also visually examine the confusion matrix and derive the same conclusion. This experiment shows that the reconstructed hypergraph crucially recovers a great amount of higher-order cluster structures in the original hypergraph.

## F.7 RUNNING TIME ANALYSIS

We have claimed that the clique sampler's complexity is close to $O(|\mathcal{M}|) + O(|\mathcal{E}|) = O(|\mathcal{M}|)$ in practice. Here we check this by asymptotic running time. Both the clique sampler and the hyperedge classifier are tested. For $p \in [30, 100]$, We sample $p\%$ hyperedges from DBLP and record the CPU time for running both modules of SHyRe-motif. The result is plot in Fig.11. It shows that both the total CPU time and the number of maximal cliques are roughly linear to the data usage (size), which verifies our claim. Fig.12 reports statistics for all methods.

Notice that among all baselines, only HyperSAGNN is suitable for running on GPUs. Other baselines run on CPUs. We can see that HyperSAGNN still runs slower than or on par with SHyRe in general. There are two main reasons for this. First, the original HyperSAGNN does not have any procedure for generating hyperedge candidates. Therefore, we have to adapt it so that it actually shares the same maximal clique algorithm with SHyRe. It also took a portion of time to sample the cliques from maximal cliques. Second, similar to DeepWalk, HyperSAGNN's best-performed variant relies on random walks sampling to generate initial node features, which also takes extensive time.

## F.8 ABLATION STUDY ON CLIQUE SAMPLER

One might argue that the optimization of the clique sampler (Sec.4.2.2) appears complex: can we adopt simpler heuristics for sampling, abandoning the notion of $r_{n,k}$'s? We study this via ablations.

We test three sampling heuristics to replace the clique sampler. 1. **"random"**: we sample $\beta$ cliques from the projection as candidates. While it is hard to achieve strict uniformness khorvash (2009), we approximate this by growing a clique from a random node and stopping when the clique reaches

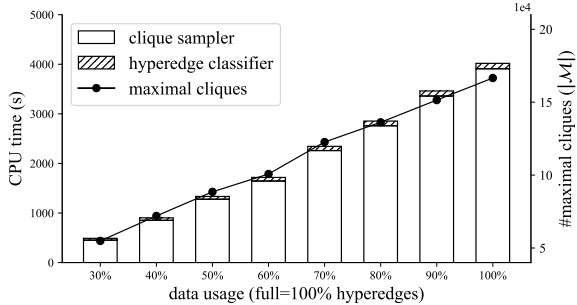

Figure 11: Asymptotic running time of SHyRe-motif on DBLP. The bar plot is aligned with the left y-axis, the line plot with the right. We can observe that both the total CPU time and the number of maximal cliques are roughly linear to the data usage (size).

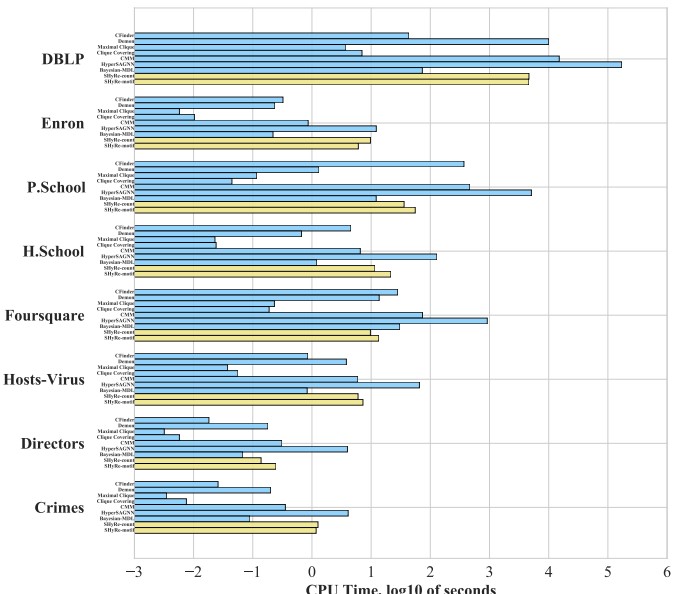

Figure 12: Comparison of running time. Notice that Bayesian-MDL's is written in C++, CMM in Matlab, and all other methods in Python.

a random size; 2.**"small"**: we sample $\beta$ cliques of sizes 1 and 2 (*i.e.* nodes and edges); 3.**"head & tail"**: we sample $\beta$ cliques from all cliques of sizes 1 and 2 as well as maximal cliques.

Fig.13 compares the efficacy in the sampling stage on Enron dataset. It shows that our clique sampler significantly outperforms all heuristics and so it cannot be replaced. Also, the the great alignment between the training curve and query curve means our clique sampler generalizes well. We further report reconstruction performance on 3 datasets in Table 10, which also confirms this point.

|  | DBLP | Hosts-Virus | Enron |
|---|---|---|---|
| original (SHyRe-motif) | 81.19±0.02 | 45.16±0.55 | 16.02±0.35 |
| ablation: "random" | 0.17±0.00 | 0.00±0.00 | 0.54±0.49 |
| ablation: "small" | 1.12±0.52 | 1.38±0.70 | 8.57±0.89 |
| ablation: "head & tail" | 27.42±0.54 | 29.92±0.54 | 11.99±0.10 |

Table 10: Ablation study: comparing the performance obtained by replacing the clique sampler with simpler heuristics for sampling.

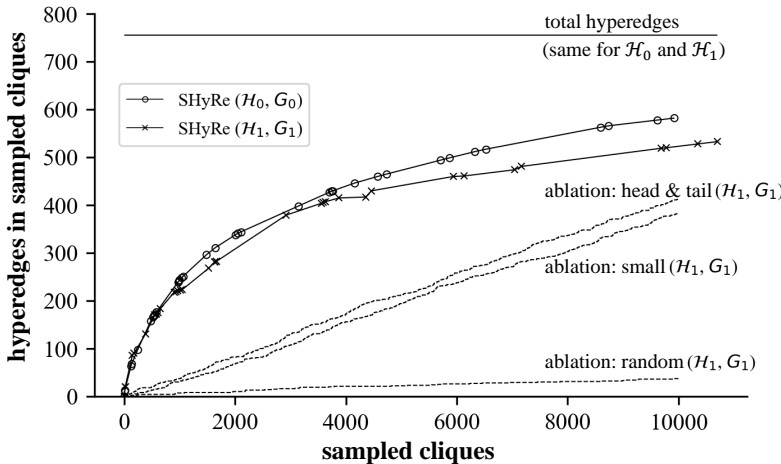

Figure 13: Ablation studies on the clique sampler. Each marker is an iteration in Algo. 1. The alignment between the two SHyRe curves shows that our clique sampler has the best generalizability.

### F.9    TASK EXTENSION I: USING EDGE MULTIPLICITIES

Throughout this work, we do not assume that the projected graph has edge multiplicities. Relying on edge multiplicities addresses a simpler version of the problem which might limit its applicability. That said, some applications may come with edge multiplicity information, and it is important to understand what is possible in this more tractable case. Here we provide an effective unsupervised method as a foundation for further work.

The multiplicity of an edge $(u, v)$ is the number of hyperedges containing both $u$ and $v$. It is not hard to show that knowledge of the edge multiplicities does not suffice to allow perfect reconstruction, and so we still must choose from among a set of available cliques to form hyperedges. In doing this with multiplicity information, we need to ensure that the cliques we select add up to the given edges multiplicities. We do this by repeatedly finding maximal cliques, removing them, and reducing the multiplicities of their edges by 1. We find that an effective heuristic is to select maximal cliques that have large size and small average edge multiplicities (combining these for example using a weighted sum).

Table 11 gives the performance on the datasets we study. We can see that with edge multiplicities our unsupervised baseline outperforms all the methods not using edge multiplicities on most datasets, showing the power of this additional information. The performance, however, is still far from perfect, and we leave the study of this interesting extension to future work.

| DBLP | Enron | P.School | H.School |
|---|---|---|---|
| 82.75 (+1.56) | 19.79 (+3.77) | 10.46 (-32.60) | 19.30 (-35.56) |
| **Foursquare** | **Hosts-Virus** | **Directors** | **Crimes** |
| 83.91 (+10.35) | 67.86 (+19.01) | 100.0 (+0.00) | 80.47 (+1.20) |

Table 11: Performance of the proposed baseline using available edge multiplicities. In parenthesis reported the increment against the best-performed method not using edge multiplicities (cr. Table 3).

### F.10    TASK EXTENSION II: RECONSTRUCTING FROM NODE-DEGREE-PRESERVING PROJECTION

Formulated in Eq.(3) of Kumar et al. (2020b), node-degree preserving projection is a novel projection method proposed in recent years that can preserve node degrees in the original hypergraph. It innovatively achieves this by scaling down the weight of each projected edge by a factor of $(|E| - 1)$, where $|E|$ is the size of the corresponding hyperedge.

Our reconstruction method can naturally extend to this novel type of projection. The reason is that our method has solved a strictly harder version of the reconstruction problem: our problem setting does not require the projected graph to have edge multiplicities (weight). In comparison, node-degree preserving projection not only produces the same set of edges as our projection does, but also provides edge weights. More importantly, node-degree-preserving projection (as its name suggests) provides the degree of each node in the original hypergraph, which is a useful piece of information in reconstruction. This information can be seamlessly integrated into our framework, as follows.

For each target clique:

1. We obtain the degree of each of its node in the original hypergraph (by computing that node's degree in the node-degree-preserving projection).
2. We compute the min and average of this new type of "node degree" for all nodes in the target clique.
3. We append the two features obtained in the last step to the "count" feature vector used in SHyRe-count.

This gives us a customized SHyRe-count model for node-degree-preserving projection. We train and test this customized SHyRe-count model on all the 8 datasets we used. The results are presented in Table 12. We observe that SHyRe's reconstruction performance on every dataset's node-degree-preserving projection is better than its performance on the regular projection used throughout this paper. These results validate our claim that our reconstruction method can naturally extend to the novel node-degree-preserving projection.

| DBLP | Enron | P.School | H.School |
|---|---|---|---|
| 83.54 (+2.36) | 14.30 (+0.80) | 44.30 (+1.70) | 55.43 (+0.87) |
| **Foursquare** | **Hosts-Virus** | **Directors** | **Crimes** |
| 76.68 (+2.08) | 50.63 (+1.78) | 100.0 (+0.00) | 87.78 (+8.60) |

Table 12: Performance of our SHyRe-count method adapted for node-degree-preserving projection. Numbers in the parenthesis are the increment over the performance of SHyRe-count on the original projections (cross referecing Table 3).

### F.11 STORAGE COMPARISON

A side bonus of having a reconstructed hypergraph versus a projected graph is that the former typically requires much less storage space. As a mini-study, we compare the storage of each hypergraph, its projected graph, and its reconstruction generated by SHyRe-count. We use the unified data structure of a nested array to represent the list of edges/hyperedges. Each node is indexed by an int64. Fig.14 visualizes the results. We see that the reconstructions take 1 to 2 orders of magnitude less storage space than the projected graphs and are closest to the originals.

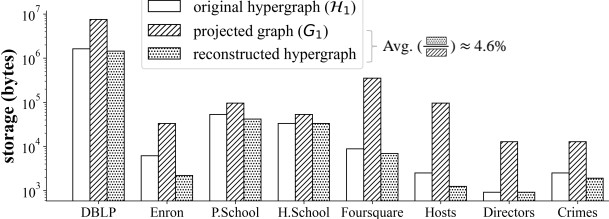

Figure 14: Comparing the storage of original hypergraphs, projected graphs and reconstructed hypergraphs. Each hypergraph/graph is stored as a nested array with `int64` node indexes. Over the 8 datasets on average, a reconstructed hypergraph takes only $4.6\%$ the storage of a projected graph.

### F.12 MORE EXPERIMENTAL RESULTS

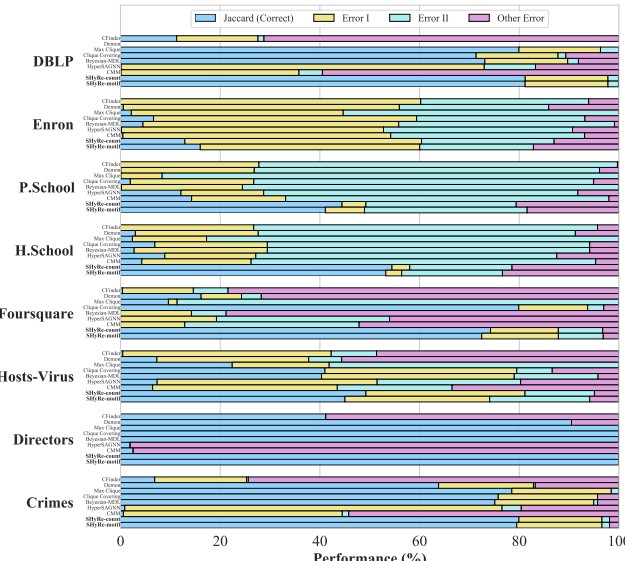

Figure 15: Distribution of the $r_{n,k}$'s of the optimal sampling strategy on all datasets. Cells of the first column and the diagonal seem to have better chance to be included, but there also exist exceptions, such as Enron (second column), P.School and H.School (second to fourth columns), etc.

Figure 16: Partitioned Performance of all methods on all datasets. Recall that the Errors I and II are mistakes made by Max Clique (Def.1). Other methods may make mistakes that Max Clique does not make, which are counted as "Other Error". We can see that SHyRe reduces more Errors I and II than other baselines do.

# G   MORE DISCUSSION ON LIMITATIONS AND FUTURE WORK

## G.1   DIRECTION I: OPTIMIZING SAMPLING RATES USING DOWNSTREAM CLASSIFICATION LOSS

One limitation in the presented reconstruction method is that it does not adjust the sampling rate (*i.e.*, the $r_{n,k}$'s) based on the downstream classifier's performance on cliques sampled at different $(n, k)$'s. In other words, because we have a constraint budget, ideally we should allocate more sampling budget to sample the types of cliques on which the downstream classifier excels at making judgement. However, under the current framework, our framework is not optimizable in an end-to-end fashion using gradient descent. Formally speaking, the current reconstruction framework is essentially optimizing the following objective $f(r, \theta)$:

$$f(r, \theta) = \max_r \min_\theta \mathbb{E}_{D \sim p(\mathcal{D}|r, \mathcal{H})} \mathcal{L}(\theta, D) \tag{2}$$

where $r$ is the sampling rates of all cells, i.e., the vector of all $r_{n,k}$'s as defined under Sec.4.2.1; $\theta$ is the parameters of our hyperedge sampler; $\mathcal{D}$ is the distribution of all possible hyperedge candidate sets that we can construct based on given sample rates $r$ and the training hypergraph $H$; $D \in \mathcal{D}$ is the specific instance of hyperedge candidate set that we end up sampling and constructing.

It would obviously be highly desirable if we can optimize $f$ with respect to both $r$ and $\theta$ using gradient descent. Fixing $r$, to optimize $f$ with respect to $\theta$ via gradient descent is straightforward, which is exactly the job done by our current hyperedge classifier. However, optimizing $f$ with respect to both $r$ is much trickier. Notice that $f$ depends on $r$ through $D$ which essentially is a discrete data

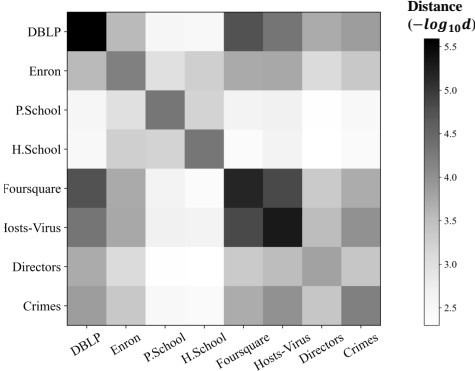

Figure 17: Pairwise distance between $\rho(n, k)$ distribution of all datasets using mean squared difference of all cell values (after alignment). The distance matrix obtained is shown above. The diagonal cell is the darkest among its row (or column).

| DBLP | Enron | P.School | H.School |
|---|---|---|---|
| $1.0E6$, ($\gg 6.0E10$) | $1.0E3$, $6.9E4$ | $3.5E5$, $2.6E6$ | $6.0E4$, $8.4E5$ |
| **Foursquare** | **Hosts-Virus** | **Directors** | **Crimes** |
| $2.0E4$, ($\gg 1.1E12$) | $6.0E3$, ($\gg 2.2E12$) | $800$, $4.5E5$ | $1.0E3$, $2.9E5$ |

Table 13: Optimal clique sampling number $\beta$ and total number of cliques $|\mathcal{U}|$. "$\gg$" appears if the dataset contains too many cliques to be enumerated by our machine in 24 hours, in which case a conservative lower bound is estimated instead.

representation, i.e., a set of cliques. This blocks the gradient flow and can could be very challenging to address.

We propose a simple workaround here which is to alternating the optimization of $r$ and $\theta$, in two phases:

- In phase 1, we fix the parameters $\theta$ of our hyperedge classifier, and optimize the parameter $r$ of our clique sampler using Algorithm 1 in the paper;

- In phase 2, we fix the parameter $r$ of our clique sampler, and optimize the parameters $\theta$ of our hyperedge classifier by standard supervised learning.

The proposed solution above works in a similar manner as k-means. However, it is not guaranteed that the global minimum will be reached. We consider this as a very intriguing direction to explore in the future.

## G.2 USING ATTRIBUTES IN RECONSTRUCTION

Another direction we may consider is the usage of node attributes in the learning-based framework. Our current method targets the most difficult version of the reconstruction problem: we don't assume nodes or edges to have attributes, and the reconstruction is purely based on leveraging structural information in the projected graph. It would be interesting to think about how we can effectively integrating node attributes into the picture, since in practice we may be able to collect some information about nodes in the projected graph. This is a nontrivial problem to research though, because the projected graph has special clique structures (and with max cliques that we have preprocessed). Therefore, how to conduct graph learning most effectively on these type of clique structures could be something worth exploring in the future.

