# Appendix

## A  REPRODUCIBILITY

Our code and data can be downloaded from `https://anonymous.4open.science/r/supervised_hypergraph_reconstruction-FD0B/README.md`.

## B  PROOFS AND ADDITIONAL DISCUSSIONS FOR SEC.3

### B.1  PROOF OF THEOREM 1

*Proof.* From Def.2, it suffices to show that for every $C$ that is not a maximal clique, $C$ is not in $\mathcal{E}$. This holds because in that case $C$ has to be the proper subset of some maximal clique $C' \in \mathcal{E}$, but since $H$ is Sperner, $C$ cannot be a hyperedge. $\square$

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

The *Expectation of the Cardinality* of a RFS is denoted by $\mathbb{E}^c$ such that $\mathbb{E}^c[A] = \mathbb{E}_{a \in \mathcal{S}(A)}|a|$. With these ready, we have the following propositions that hold true for RFS $A$ and $B$:

(i) $\mathbb{E}^c[A \cup B] = \mathbb{E}^c[B \cup A]$

(ii) $\mathbb{E}^c[A \cup B] = \mathbb{E}^c[A \backslash B] + \mathbb{E}^c[B]$;

(iii) $\mathbb{E}^c[A \cup B] \geq \mathbb{E}^c[A], \mathbb{E}^c[A \cup B] \geq \mathbb{E}^c[B]$;