# OpenReview forum: "From Graphs to Hypergraphs: Hypergraph Projection and its Reconstruction"
_ICLR.cc/2024/Conference — ICLR 2024 poster_

### Official Review · Reviewer_U5QZ · 2023-10-31

**Soundness:** 2 fair
**Presentation:** 2 fair
**Contribution:** 2 fair
**Rating:** 6
**Confidence:** 4

**Summary:**

The authors propose a method of constructing a hypergraph from a given graph. The proposed method includes analyzing cliques present in the graph and classifying them into potential hyperedges. The proposed method is compared against baselines, and the results show superior performance of the proposed method.

**Strengths:**

1. The problem is very important. Many times, getting hypergraph-based representation is hard. The proposed approach can be used to convert a graph-based representation to an underlying hypergraph.
2. The proposed approach builds on the clique finding algorithm, a known approach for finding potential hyperedges. Its key challenges, such as predicting hyperedges that are a complete subset of larger hyperedges, computational challenges, etc., are tackled well.
3. The results show that the proposed method outperforms the baseline approaches.

**Weaknesses:**

1. The paper lacks a comparison of their method with several other works in the domain. I have listed them in the Questions section.
2. Writing lacks a clear outline of the contributions. In the proposed pipeline, what is already proposed vs. what is novel, is not clear.
3. The problem and solution are discussed in detail, but the experiments and results are not justified. What properties of a hypergraph make the proposed algorithm more suitable? On some datasets. The baseline methods demonstrate comparable performance; what is the reason behind that?
4. Other weaknesses are asked in the Questions section.

Minor typos (does not affect the rating):
Page 3: as well as and

**Questions:**

1. Conditions stated in Theorem 1 are explained as necessary conditions to reconstruct a hypergraph from a given projection using the max-clique algorithm. But are these conditions sufficient? Is there any theoretical justification for that?
2. The clique-sampler tries to sample for as many as possible hyperedges. Will it not affect the size of hyperedges being predicted?
3. The performance of the proposed method is not very different from the baselines on hypergraphs where the average hyperedge size is large. Is there any justification for that? Also, from the predicted hyperedges, is it possible to provide a stratified evaluation to understand how the proposed method works on large vs small hyperedges?
4. Can you think of a real-world scenario where converting a graph to a hypergraph (where we know that there are underlying group interactions) is essential, and this approach can help?
5. Apart from clique-based projections, there are other hypergraph projection methods, such as the node-degree-preserving method[4]. Can this approach recover hyperedges from such projected graphs?


6. I request the authors to provide justification for the following.
a. In the Introduction section, whether the hyperedges are observable or not actually depends on the objective of experiments conducted on the system. For example, protein-protein, gene-gene interactions have semantics defined for pairwise interactions where protein complexes are inherently super-dyadic relations of proteins, and experiments observe them in the protein complex form [1].
b. Unpublished: True, author-author interaction datasets do not provide the underlying hypergraph structure, but one should go with other bibliographic datasets such as AMiner[2] to get a complete view of the system.
c. Relevance to the hyperedge prediction problem: The problem of hyperedge prediction is actually relevant. The hyperedge prediction methods consume a hypergraph as input, and the right way to see it is every graph is also a hypergraph (more precisely, 2-uniform hypergraph), and the methods [3] where a set of candidate hyperedges is not used can identify underlying hyperedges. Especially when you are using the sample hypergraphs from a given domain, methods like HPRA can generate new hyperedges using the known hyperedge degree distribution.


[1] "Hypergraphs and cellular networks." PLoS computational biology 5.5 (2009): e1000385.
[2] https://www.arnetminer.org/
[3] "HPRA: Hyperedge prediction using resource allocation." Proceedings of the 12th ACM Conference on Web Science. 2020.
[4]  "Hypergraph clustering by iteratively reweighted modularity maximization." Applied Network Science 5.1 (2020): 1-22.

---

> ### Author Response · Authors · 2023-11-19
> **Response by Authors (1/2)**
>
> We thank the reviewer for the feedback. We have revised our paper following the reviewer’s suggestions. Here are our answers to the questions.
>
>
>
> **Question 1**
>
> Conditions stated in Theorem 1 are stated as both necessary and sufficient conditions, as indicated by the “if and only if” in Theorem 1. If the reviewer is mainly concerned about the text below Theorem 1 that explains its significance, we have added the phrase ``necessary and sufficient” to avoid confusion. We have also updated our proof in Appendix B.1 to highlight that both directions are true.
>
>
> **Question 2**
>
> The clique-sampler indeed samples as many hyperedges as possible since it needs to maximize its utility objective under the budget constraint.  We design the sampling mechanism, however, to positively affect the size of hyperedges, i.e., making their sizes diverse and in line with the distribution in the training hypergraph.  This is accomplished by our Algo.1: Appendix F.2.1 elaborates on this, with additional numerical results obtained on DBLP; Table 7 is also very helpful as it examines more properties of reconstructions. Below is a short summary:
>
> In each iteration, Algo. 1 tends to sample cliques from a different column of Figure 4(a)’s heatmap.  Cliques of the same size have diminishing sampling gain, due to the overlapping of $\mathcal{E}_{n,k}$’s that have the same $k$. Meanwhile, cliques of different sizes remain unaffected in each sampling iteration. This inherent exclusivity encourages more diverse hyperedge sizes in sampling, which reduces the chance of a skewed size distribution in the reconstruction result. Of course, ultimately the sampling is still guided by the distribution of hyperedges in the training hypergraph.
>
> **Question 3 & Weakness 3**
>
> We have added a new section, Appendix F.3, to answer this question systematically through a linear regression on the performance gap.
>
> As a short summary, we found that (1) our advantage over baselines is most prominent on dense hypergraphs, because dense hypergraphs are usually hardest to reconstruct; (2) our advantage is relatively less prominent on datasets with larger hyperedges, because baselines like max cliques, clique covering, and Bayesian-MDL naturally favor large hyperedge in reconstruction. In these cases, the baseline’s assumption is aligned with the dataset property.
> Please read F.3 for more details.
>
> **Question 4 & Question 6a**
>
> As introduced in Section 5.4 and elaborated in Appendix F.4, a realistic use case we have demonstrated is multiprotein complex reconstruction. This is a widely-studied application that naturally fits the motivation for hypergraph reconstruction, since it is a case where some of the most readily available measurement technologies can only record pairwise interactions in a system that is fundamentally composed of hyperedges.  In particular, we understand the reviewer’s point that it can be an experimental choice as to whether one should detect protein interactions as pairs or groups, but as we elaborate in Appendix F.4, pair-based detection methods such as Y2H are much more accessible and faster than group-based detection methods such as TAP-MS. As a result, a large amount of the data so far generated from protein interaction studies have come from Y2H screenings. This often leads to a realistic demand for researchers to recover groups (hyperedges) from pair detection results. Please refer to Appendix F.4 for more details.
>
>
> **Question 5**
>
> Yes. Our approach can also recover hyperedges from the novel node-degree-preserving projection, with even better performance than under this paper’s harsh setting. We have added a section in Appendix F.10 to elaborate on this.
>
>
>
>
> **Question 6b**
>
> It would certainly be most ideal if one could cross-reference other bibliographic datasets. However, since almost all popular co-authorship networks (e.g. obgl-arxiv, Arxiv Hep-Th) have anonymized node identifiers for privacy issues, it is often very difficult to even figure out the actual author that each node represents. In the meantime, directly reaching out to data publishers to request for the source takes significant time and effort, and usually still runs into the same privacy issues. In these cases, hypergraph reconstruction becomes a much more economic and feasible choice.
>
> **(to continue)**

---

> ### Author Response · Authors · 2023-11-19
> **Response by Authors (2/2)**
>
> **(Continued)**
>
> **Question 6c**
>
> We agree with the reviewer that the suggested HPRA method is relevant, and we have added it as a baseline for comparison (see the updated Table 3).
>
> HPRA is a novel method for hyperedge prediction, and it addresses the huge search space of hyperedges very well. It also has the advantage of being lightweight, interpretable, and requiring no tuning at all. As Table 3 shows, the HPRA is a strong baseline that achieves competitive performance among other baselines. However, we still observe a gap in its performance with our proposed method for hypergraph reconstruction, for two reasons:
>
> -  HPRA does not reconstruct all hyperedges from scratch. Instead, it requires at least 80% - 90 % of the hyperedges to already exist in the test hypergraph (See “Evaluation of HPRA” in Section 4 of its paper, where they use 5-fold or 10-fold split). When we treat edges in the projected graph as 2-uniform hyperedges, this violates the underlying assumption that the hypergraph to predict (reconstruct) is ``almost’’ complete, leading to two issues:
>
>     -  Forcing HPRA to reconstruct from scratch causes its errors in the early stage of reconstruction to quickly accumulate. Notice that HPRA’s prediction of each new hyperedge depends on its previous predictions, i.e., it is self-regressing.
>
>     - Treating the hypergraph as a 2-uniform graph significantly alters the assumed hypergraph structure. In other words, it essentially degrades the hypergraph predictions into samplings from the landing probabilities of random walks on the projected graph.
>
> - HPRA uses the training hypergraph (“observed hyperedges”) in a limited way. It mainly learns about the size distribution of hyperedges from the training hypergraph. In comparison, our method not only learns about size distributions (via $\rho(n,k)$’s), but more importantly learns from the fine-grained statistics about hyperedges’ locations, as well as the location’s intricate correlation with their surrounding structures.
>
>
>
> **Weakness 1**
>
> Please see our answers to Question 5 and 6.
>
> **Weakness 2**
>
> We have followed the reviewer's suggestion to add a paragraph in Introduction, which summarizes our contributions in this paper.
>
> We summarize the contributions of the proposed pipeline as follows. In parenthesis, we also paste excerpts from the text of our paper where the novelty and originality of the design are emphasized. Our contributions of the proposed pipeline are:
>  - The entire paradigm of learning-based hypergraph reconstruction (Sec.4, outline: “This section will introduce a new learning-based hypergraph reconstruction paradigm. The idea is that …”)
> -  The idea of reconstructing hyperedges by combining budgeted sampling and clique classification (Sec. 4.2: “the novel idea here is that …”)
> - The entire design of the $\rho(n,k)$ statistic; (Sec. 4.2.1: “We start by introducing an important statistic that we found …”)
> - The entire clique sampler, including the objective, optimization algorithm, etc. (Sec. 4.2.2: “we create a clique sampler …”)
> - The entire design of the hyperedge classifier based on “clique motifs”
> - All count features used in the hyperedge classifier, except features 2,3,7, are new contributions of our paper. Features 2,3,7 are very commonly used count features.
>
> To further address the reviewer’s concern, we have revised our writing to reflect the novelty of the last two bullet points above.
>
> **Weakness 3**
>
> Please see our answer to Question 3
>
> **Weakness 4**
>
> Please see our answers to the Questions.
>
> We have also fixed the typo, and we appreciate the reviewer for pointing it out.

---

> ### Author Response · Authors · 2023-11-21
>
> Dear Reviewer,
>
> We're appreciative of your feedback on our work! As the discussion period draws to a close, we hope to respectfully check if you still have any concerns or suggestions. We would greatly value your insights on our response and updates!
>
> Warmest regards,
>
> Authors of Paper 4521

---

> > ### Comment · Reviewer_U5QZ · 2023-11-21
> > **Thank you for the response.**
> >
> > Thank you for providing clarification to all my comments. My suggestion is to revise the manuscript to reflect these changes (up to the authors what they feel is the best way). I have revised my rating. All the best!

---

> > > ### Author Response · Authors · 2023-11-21
> > >
> > > Thank you for raising the rating! We will follow your suggestion to revise the manuscript and reflect the changes.

---

### Official Review · Reviewer_rZdL · 2023-10-31

**Soundness:** 4 excellent
**Presentation:** 4 excellent
**Contribution:** 4 excellent
**Rating:** 8
**Confidence:** 5

**Summary:**

This paper studies the task of recovering a hypergraph from its projected graph (i.e., the graph formed by forming an edge between any two nodes that are a part of the same hyper-edge).

The first main contribution is a set of theoretical results stating when it is possible to reconstruct the hyper-edges based on the maximal cliques of the projected graph, G. In particular, the authors identify two necessary and sufficient conditions for the recovery of the hypergraph from G: (1) the lack of any "nested" hyper-edges, and (2) the lack of any "uncovered triangles" (Thm 1). Moreover, when (2) is not satisfied, the reconstruction accuracy using maximal cliques can be exponentially small in the number of hyper-edges (Thm 3).

The second main contribution is a learning-based approach that leverages side information in the form of same-domain hypergraphs to reconstruct a hypergraph from its projection. The method follows a 4-step procedure. First, the distribution of hyper-edges within maximal cliques of G is computed for the training data (the known hypergraph of the same domain); these are denoted by the $\rho(n,k)$'s. Using this information, cliques in G are sampled  in a way that is optimized for the $\rho(n,k)$'s and a query budget (since the total search space is very large). Finally, cliques are classified as hyperedges or not based on the local structure of the cliques with respect to the surrounding graph. The methodology is assessed on a variety of datasets with favorable results.

**Strengths:**

Reconstructing hypergraphs from their graph projections is an important task in several domains, and this is summarized very well by the authors in the introduction. This task has received little to no attention previously, making the fundamental analysis in this paper quite valuable.

The authors' contributions are substantive and fundamental, spanning both theory and practical implementations. The motivation for studying cliques is clear, and the authors derive a nice, fundamental characterization for general hypergraph recovery using maximal cliques. The authors' development of a scalable, learning-based approach to overcome the shortcomings of an approach without side information serves as a nice, practical complement to the theoretical results.

**Weaknesses:**

The approaches outlined in the paper are somewhat basic (which itself is not a weakness, as the problem is novel and the contributions fundamental). What are potential future directions for the design of possibly more sophisticated methods? What other types of information could be taken into account to improve guarantees for hypergraph reconstruction? Some discussion of these and related questions would be quite valuable.

**Questions:**

- Concerning the characterization of "conformal" in Theorem 2, it might be worth describing the explicit connection between uncovered triangles and hyper-edge reconstruction. That is, the triangle would induce a 3-clique among the "points" of the triangle, which would not be part of a hyper-edge if the triangle is uncovered.
- Are there any natural strategies you'd expect to perform better than looking at maximal cliques, for the case of no side information?
- The notation $\mathbb{E}^c$ (denoting expected cardinality) is a bit nonstandard. Perhaps just write $\mathbb{E}[|S|]$ for the expected size of a set S.

---

> ### Author Response · Authors · 2023-11-19
> **Response by Authors**
>
> We thank the reviewer for the feedback. We have revised our paper following the reviewer’s suggestions. Here are our answers to the questions.
>
> **Weakness**
>
> We agree with the reviewer to discuss more about future directions. We have followed the advice to add an entire section, “More Discussions on Limitations and Future Work”, in Appendix G.
>
> **Question 1**
>
> We agree with the reviewer. We have added to our manuscript this explicit connection between uncovered triangles and hyper-edge reconstruction – see highlighted text under Theorem 2.  We would also like to point to the caption of Figure 2, where we presented a concrete example of how the uncovered triangle fails hyperedge reconstruction.
>
> **Question 2**
>
> We have counted on the “(Edge) Clique Covering” baseline to outperform the “Max Cliques” baseline. As introduced in Appendix C.2, “Clique Covering” is a classical algorithmic problem that aims to use the least number of cliques to cover all the edges of the projected graph.  This is a very natural baseline for reconstructing hyperedges. This is also the only baseline that strictly enforces the principle of parsimony --  a popular rule of thumb in science. Therefore, we thought it should perform better than max cliques.
> In practice, our experimental results in Table 3 show that “Clique Covering” does outperform “Max Cliques” on 5 out of the 8 datasets.  Meanwhile, however, it also performs very badly on Enron, P.School, and H. School. We conclude that the principle of parsimony is useful in some cases of reconstruction, but not (even close to) good in many others. In fact, we think there may not exist an (unsupervised) strategy that is universally applicable to reconstructing any real-world hypergraphs, which necessitates the introduction of a learning-based reconstruction method.
>
> **Question 3**
>
> We have followed the advice to fix all notations of E^c in our revised manuscript.
>
> One direction is to consider propagating the error signal (e.g., loss gradient) from the hyperedge classifier back to the clique sampler. We have appended an additional section, Appendix G, to formulate the challenge in this interesting future direction, as well as a conjectured solution.
>
> Another direction we may consider is the usage of attributes in the learning-based framework. Our current method targets the most difficult version of the reconstruction problem: we don’t assume nodes or edges to have attributes, and the reconstruction is purely based on leveraging structural information in the projected graph. It would be interesting to think about how we can effectively integrate node attributes into the picture. This is nontrivial though because the projected graph has special clique structures (and with max cliques that we have preprocessed). Therefore, how to conduct graph learning most effectively on these type of clique structures could be something worth exploring.

---

> ### Author Response · Authors · 2023-11-21
>
> Dear Reviewer,
>
> We're appreciative of your feedback on our work! As the discussion period draws to a close, we hope to respectfully check if you still have any concerns or suggestions. We would greatly value your insights on our response and updates!
>
> Warmest regards,
> Authors of Paper 4521

---

### Official Review · Reviewer_zeZp · 2023-11-01

**Soundness:** 3 good
**Presentation:** 3 good
**Contribution:** 3 good
**Rating:** 8
**Confidence:** 3

**Summary:**

This paper investigates the reconstruction of a hypergraph from a projected graph. The authors propose a learning-based hypergraph reconstruction method based on their observation of the distributions of hyperedges within maximal cliques. They utilize a clique sampler and a hyperedge classifier to reconstruct hypergraphs.

The authors evaluate their method for hypergraph reconstruction and downstream tasks using the reconstructed hypergraphs. They show that their method outperforms existing methods for hypergraph reconstruction. Using hypergraphs reconstructed by their method improves performance on downstream tasks compared to using projected graphs.

**Strengths:**

1. This paper is well-written and easy to understand.

2. The proposed method effectively address problems with appropriate approaches.

3. It seems to provide a foundation for the underexplored problem of hypergraph reconstruction from a projected graph.

4. The experimental setup for reconstruction is well-designed and the proposed method yields good performance.

**Weaknesses:**

1. I'm uncertain about the necessity of addressing this problem.

2. It would be advantageous to incorporate supplementary experiments to illustrate the benefits of utilizing reconstructed hypergraphs. Although the experiments show the proposed method's strong performance in reconstructing hypergraphs, its impact on downstream tasks remains unclear.

3. In experiments for link prediction (In F.4), the authors append several structural features of hyperedges to the final link embeddings. However, in cases of projected graphs, additional structural features cannot be used. Is it a fair comparision?

**Questions:**

1. Could the performance improvement in experiments on link prediction have originated from the use of additional structural information?

2. Could the authors demonstrate the benefits of reconstructed hypergraphs through additional experiments?

---

> ### Author Response · Authors · 2023-11-19
> **Response by Authors**
>
> We thank the reviewer for the feedback. We have revised our paper following the reviewer’s suggestions. Here are our answers to the questions.
>
> **Weakness 1**
>
> First, a foundational motivation to study the problem of hypergraph reconstruction is that we have very limited understanding about the consequences of hypergraph projection, a process that implicitly happens very frequently in network analysis. This is a source of concern both conceptually and in applications because we don’t know what we miss when we drastically simplify complex systems involving hypergraphs in this way. We address this question in two fundamental ways: (i) We begin by presenting several theorems to mathematically characterize the consequences of hypergraph projection in general terms (we consider these to be some of our most important findings); and (ii) we study the problem that is the inverse of hypergraph projection, which is hypergraph reconstruction.
>
> Regarding the actual hypergraph reconstruction problem, it has the following benefits:
>
> - The reconstructed hypergraphs themselves are very useful data representations, because they crucially tell us which group of nodes in a projected graph are really interacting with each other at a time. Therefore, the reconstructed hypergraphs can facilitate scientific analysis that are infeasible or less accurate in projected graphs. Examples include analysis of protein complexes and social interactions; in both of these cases, as we discuss in the paper, natural ways of measuring the system produce graph representations for what are in fact hypergraph structures.
> -  The reconstructed hypergraphs also facilitate downstream tasks, though they do not exist for the sole purpose of improving any one specific downstream task. We demonstrate this through the link prediction task. Please read more in our answer to “Weakness 3 & Question 1”.
> -  Even the instances of hypergraph reconstruction that yield poor performance results have value, since they help us identify the classes of hypergraphs for which projection risks causing the greatest amount of information loss.  For example, one interesting finding in our experiment is that hypergraphs with large hyperedges are relatively safer to project, while it is more important to avoid projecting hypergraphs with high node degrees.
>
>
>
> **Weakness 2 & Question 2**
>
> We have followed the reviewer’s suggestion to add another use case in Appendix F.6, showing reconstructed hypergraph’s benefit in the classical task of node clustering. In particular, we show that the hypergraph reconstruction crucially recovers much information about the higher-order cluster structures in a social interaction network.
> We believe that the benefits of reconstructed hypergraph in downstream tasks have now been well substantiated by the three concrete use cases: node ranking (F.4), link prediction (F.5), and node clustering (F.6).
>
>
>
> **Weakness 3 & Question 1**
>
> The improvement on link prediction in Appendix F.4 does originate from the use of additional structural information. However, this is exactly what we have wanted to demonstrate. Without a hypergraph reconstruction method, the classifier for link prediction is unable to use higher-order structural features in the (testing) projected graph. This limitation exists despite the availability of a training hypergraph together as input. This is because there is no mapping between nodes in the testing projected graph and nodes in the training hypergraph. Therefore, we can’t use training hypergraph to construct structural features for nodes in the tested projected graph.
>
> Hypergraph reconstruction, on the other hand, creates a mechanism to use information in the training hypergraph when the task is defined on the testing projected graph (by first reconstructing the testing hypergraph).  This is an important sense in which hypergraph reconstruction — by providing access to an inferred hypergraph structure — unlocks the potential for using available side information that is not directly compatible with graph structures.
>
> We also hope to emphasize that we do not intend to compete against the current state-of-the-art method for link prediction. The main purpose of this experiment is to show that the reconstructed hypergraph contains structural information that is useful and handy for many analysis tasks, including link prediction.

---

> > ### Comment · Reviewer_zeZp · 2023-11-23
> >
> > I appreciate the authors for answering my questions. All of my concerns have been addressed.
> > I will raise my score to 8.

---

> ### Author Response · Authors · 2023-11-21
>
> Dear Reviewer,
>
> We're appreciative of your feedback on our work! As the discussion period draws to a close, we hope to respectfully check if you still have any concerns or suggestions. We would greatly value your insights on our response and updates!
>
> Warmest regards,
> Authors of Paper 4521

---

> > ### Comment · Reviewer_rZdL · 2023-11-22
> >
> > Thanks to the authors for addressing my comments and updating the manuscript. I shall maintain my score on this work.

---

### Official Review · Reviewer_EEWq · 2023-11-02

**Soundness:** 3 good
**Presentation:** 4 excellent
**Contribution:** 3 good
**Rating:** 8
**Confidence:** 5

**Summary:**

The authors (a) analyze the hardness of reconstructing a hypergraph from its clique expansion, (b) propose a supervised algorithm for hypergraph reconstruction, and (c) empirically demonstrate the effectiveness of the proposed algorithm, compared to previous approaches. The proposed algorithm begins by sampling potential hyperedges based on domain-based patterns, followed by the classification of these candidates.

**Strengths:**

S1. The proposed formulation and algorithm are a novel blend of empirical insights and theory.

S2. The difficulty of problem at hand and the approximation guarantee of the proposed algorithm are theoretically analyzed.

S3. The proposed method reconstructs hypergraphs more accurately, compared to baseline approaches, and the reconstructed hypergraphs are shown useful for downstream tasks compared to graph representations (i.e., clique expansions).

S4. The paper is exceptionally well-written.

**Weaknesses:**

W1. The graph representations (e.g., clique expansions) and hypergraph representations have been compared in many contexts, which are largely ignored in the paper (see [R1]-[R3]).
- [R1] The why, how, and when of representations for complex systems
- [R2] How Much and When Do We Need Higher-order Information in Hypergraphs? A Case Study on Hyperedge Prediction
- [R3] HNHN: Hypergraph Networks with Hyperedge Neurons

W2. The comparison between the structures of the original hypergraphs and the reconstructed ones is limited, relying on basic statistical metrics. It's important to note that hypergraph structures can exhibit variations even when their basic statistics align. To conduct a more comprehensive comparison, additional measures and analytical tools should be considered [R4].
- [R4] Mining of Real-world Hypergraphs: Patterns, Tools, and Generators

W3. The algorithm is divided into two components: sampling and classification. Notably, the classification accuracy may vary across different (n,k) combinations. However, this aspect was not considered during the formulation and theoretical analysis of the sampling part.

**Questions:**

I served as a reviewer for this submission at other conferences and had enough opportunities to (anonymously) interact with the authors. I do not have any further questions or inquiries.

---

> ### Author Response · Authors · 2023-11-19
> **Response by Authors**
>
> We thank the reviewer for the feedback. We have revised our paper following the reviewer’s suggestions. Here are our answers to the questions.
>
>
> **W1**
>
> We have followed the suggestion to add an additional subsection, Appendix C.1, which includes more extensive discussion on references R1-R3. We appreciate these references as they also help highlight the importance of our studied topic.
>
>
> **W2**
>
> We have followed the suggestion to add an additional subsection, Appendix F.2.2, to our manuscript, which reports numerical results on some of the more advanced statistics, including simplicial closure, degree distribution, singular-value distribution, density, and diameter. We will keep working on expanding these numerical comparisons.
>
> **W3**
>
> We have followed the suggestion to add a section to our manuscript as Appendix G, so as to (1) mathematically formulate the reviewer’s idea, (2) analyze the main challenge in doing so, and (3) propose an outline of our conjectured solution.
> We agree with the reviewer that the current framework has not optimized the sampling step by leveraging error signals / gradients propagated back from the downstream classification step. We acknowledge that a perfect solution to handle the blocking of gradient flow in the current framework structure is still under-explored, and that it would indeed be highly intriguing to further study this idea. We consider our current work as a foundational first step towards a long-term agenda to research the problem of hypergraph reconstruction.

---

> ### Author Response · Authors · 2023-11-21
>
> Dear Reviewer,
>
> We're appreciative of your feedback on our work! As the discussion period draws to a close, we hope to respectfully check if you still have any concerns or suggestions. We would greatly value your insights on our response and updates!
>
> Warmest regards,
> Authors of Paper 4521

---

> > ### Comment · Reviewer_EEWq · 2023-11-21
> > **Thanks**
> >
> > I appreciate the authors for addressing my concerns. I would like to maintain my positive score.

---

### Author Response · Authors · 2023-11-23
**Summary of Reviews and Discussion by Authors**

As the discussion period draws to a close, we the authors would like to thank all reviewers again for their feedback:

- Reviewer EEWq thinks this paper is "exceptionally well written" with "a novel blend of empirical insights and theory";
- Reviewer zeZp thinks this paper is "well-written", and it "provides a foundation for the underexplored problem of hypergraph reconstruction", with "well-designed experiment";
- Reviewer rZdL thinks this paper "studies an important task that receives little to no attention before", and its "contributions are substantive and fundamental, spanning both theory and practical implementations";
- Reviewer U5QZ thinks the studied problem "is very important", and that the approach "addresses key challenges appropriately"


We have made detailed responses to all reviewers' questions and concerns. We have also revised our paper to reflect all suggested refinement by reviewers. These have led to positive comments by the reviewers, in which they indicate their satisfaction.

---

### Public Comment · ~Pan_Li5 · 2023-11-24
**Thanks for the great work and some missing relevant works**

I am reading through ICLR submissions and see this work. I think this is a great work and provides a solid investigation on an interesting problem. I am writing to mention some very relevant works on graph and hypergraph relationship, and hyperedge prediction, not having been discussed in this work. Note that I emphasize this is a great work and I do not think missing such discussion disqualifies this work's own content, but I have to mention them to give the community the entire picture of this line of research.

The introduction states that "prior works Very limited work investigated implications of hypergraph projection or its reversal
(i.e. hypergraph reconstruction)...". Actually, there have been a line of studies [1][2] and even some works cited by [1][2] that investigated hypergraph spectral theory and disclosed the pros and cons of hypergraph projection. See "Unfortunately, the clique expansion (i.e., hypergraph project) method in general has two drawbacks: The spectrum for graphs used in the second step is merely quadratically optimal, while the projection step can cause a large distortion." in [2]. Theorem 3.4 in [1] directly shows using hypergraph projection to represent high-order relationship modeled by hyperedges may induce distortion.

Moreover, this work cited our paper [3] which first introduced how to use GNNs to predict triangles in the appendix. Note that [3] still works on graphs and predict triangles (3-cycles) not hyperedges. Actually, the technique (structural feature encoding) was later  indeed
generalized to predict hyperedges [4], and dynamic hyperedges [5] in temporal hypergraphs. [5] due to the complexity also relies on some subsampling to construct the training dataset. Papers that cite these works may also target some hyperedge prediction tasks. I sincerely hope the authors may give proper credits to these studies. :)

[1] Inhomogeneous Hypergraph Clustering with Applications, NeurIPS 2017.
[2]  Submodular hypergraphs: p-laplacians, cheeger inequalities and spectral clustering, ICML 2018.
[3] Distance encoding: Design provably more powerful neural networks for graph representation learning, NeurIPS 2020
[4] Principled hyperedge prediction with structural spectral features and neural networks, arxiv 2021
[5] Neural Predicting Higher-order Patterns in Temporal Networks, WWW 2022

---

> ### Public Comment · ~Yanbang_Wang1 · 2024-05-07
>
> Thank you very much for your comment.
>
> We are glad that the commenter thinks our submission is “great work” and we appreciate their support of the paper. We think these earlier papers they mention are central and strong contributions, and we will add them to the arXiv version of our paper as references.
>
> However, we also want to emphasize the ways in which these earlier papers are working on fundamentally different problems from ours. We understand that the commenter also appreciates these distinctions, and therefore noted in the comment that their existence doesn’t detract from our paper, but we wanted to help make sure that the distinctions were clear between the problems these earlier papers considered and the fundamentally different problem that our paper considers.
>
> In particular, those earlier papers are for downstream applications like hyperedge prediction and hypergraph clustering. None of them are designed to study the four fundamental questions about consequences of hypergraph projection and its inverse problem as formulated in Q1-Q4 of the Introduction to our paper. Our paper is pursuing questions in a distinct direction, in that we want to understand in general terms what information is lost via hypergraph projection, and specifically the extent to which this information can be (partially) recovered through the inverse operation of hypergraph reconstruction.  We view our answers to this distinct set of questions as the primary contribution of our work.
>
> Regarding the references provided by the commenter:
>
> For the hyperedge prediction task [3,4,5], the last paragraph in our Introduction has explained that this task is fundamentally different from the hypergraph reconstruction task studied in our work. The two tasks have different input and different output, with very different challenges to address. Still, we have referenced (but not exhausted) previous methods for hyperedge prediction, including Hyper-SAGNN, HPRA, and CMM. The subsampling technique used in [5] is to address the class imbalance problem in hyperedge type prediction, which is fundamentally different from our clique sampler that addresses the enormous search space of hyperedges.
>
> For the hypergraph clustering task [1,2], we want to mention that: (1) In [1], “spectrum for graphs used in the second step is merely quadratically optimal” is not an issue caused by hypergraph projection. Instead, it is a well-known fact with spectral clustering on all graphs. (2) [1] also does not study the “large distortion” caused by projection, but circumvents this issue through $p$-Laplacian. (3) [2]’s Thrm 3.4 does help characterize the competitive ratio of projection-based clustering under several assumptions, including linear hyperedge weight function. We believe that it belongs to the same line of research on projection-based clustering, as exemplified by our existing reference of Wolf et al. 2016. The observed distortion in projection-based clustering makes hypergraph projection and its remediation a more valuable problem to systematically study – the main topic in this work.
>
>
> Warm regards,
>
> Authors of "From Graphs to Hypergraphs: Hypergraph Projection and its Reconstruction"

---

### Meta-Review · Area_Chair_Rmeg · 2023-12-07

**Metareview:**

The paper introduces and studies the problem of reconstructing a hypergraph from a graph representation of the hypergraph where each hyperedge is replaced by a clique on its vertices. The paper studies this problem in a learning-based setting, and it proposes a supervised learning algorithm for reconstructing the hypergraph.

The reviewers appreciated the novelty of the problem formulation and the theoretical and empirical contributions of the paper. The author response addressed the reviewers' feedback. Overall, there was consensus that this work makes an interesting and valuable contribution to a well-motivated problem.

**Justification For Why Not Higher Score:**

Although the paper's contributions are solid and relevant, they may not rise to the level of a spotlight presentation.

**Justification For Why Not Lower Score:**

The paper makes novel contributions for a well-motivated problem domain.

---

### Decision · Program_Chairs · 2024-01-16

Accept (poster)